# Distinguishing mutants that resist drugs via different mechanisms by examining fitness tradeoffs

Kara Schmidlin[1,2†], Sam Apodaca[1,2†], Daphne Newell[1,2], Alexander Sastokas[1,2], Grant Kinsler[3], Kerry Geiler-Samerotte[1,2*]

[1]Biodesign Center for Mechanisms of Evolution, Arizona State University, Tempe, United States; [2]School of Life Sciences, Arizona State University, Tempe, United States; [3]Department of Bioengineering, University of Pennsylvania, Philadelphia, United States

*For correspondence:
kerry.samerotte@asu.edu

†These authors contributed equally to this work

Competing interest: The authors declare that no competing interests exist.

**Abstract** There is growing interest in designing multidrug therapies that leverage tradeoffs to combat resistance. Tradeoffs are common in evolution and occur when, for example, resistance to one drug results in sensitivity to another. Major questions remain about the extent to which tradeoffs are reliable, specifically, whether the mutants that provide resistance to a given drug all suffer similar tradeoffs. This question is difficult because the drug-resistant mutants observed in the clinic, and even those evolved in controlled laboratory settings, are often biased towards those that provide large fitness benefits. Thus, the mutations (and mechanisms) that provide drug resistance may be more diverse than current data suggests. Here, we perform evolution experiments utilizing lineage-tracking to capture a fuller spectrum of mutations that give yeast cells a fitness advantage in fluconazole, a common antifungal drug. We then quantify fitness tradeoffs for each of 774 evolved mutants across 12 environments, finding these mutants group into classes with characteristically different tradeoffs. Their unique tradeoffs may imply that each group of mutants affects fitness through different underlying mechanisms. Some of the groupings we find are surprising. For example, we find some mutants that resist single drugs do not resist their combination, while others do. And some mutants to the same gene have different tradeoffs than others. These findings, on one hand, demonstrate the difficulty in relying on consistent or intuitive tradeoffs when designing multidrug treatments. On the other hand, by demonstrating that hundreds of adaptive mutations can be reduced to a few groups with characteristic tradeoffs, our findings may yet empower multidrug strategies that leverage tradeoffs to combat resistance. More generally speaking, by grouping mutants that likely affect fitness through similar underlying mechanisms, our work guides efforts to map the phenotypic effects of mutation.

## eLife assessment

This study provides **valuable** new insights into the trade-offs associated with the evolution of drug resistance in the yeast *S. cerevisiae*, based on a solid approach to evolving and phenotyping hundreds of independent strains. The authors identify distinct phenotypic clusters, defined by their growth across defined conditions, which suggest that tradeoffs are diverse but at the same time could be limited to a few classes according to the underlying resistance mechanisms. The methodologies used align with the current state-of-the-art, and the data and analysis are **solid** as they broadly support the claims, with only a few minor weaknesses remaining after revision. This work will interest molecular biologists working on the evolution of new phenotypes and microbiologists studying multi-drug therapy.

**eLife digest** Mutations in an organism's DNA make the individual more likely to survive and reproduce in its environment, passing on its mutations to the next generation. Mutations can alter the proteins that a gene codes for in many ways. This leads to a situation where seemingly similar mutations – such as two mutations in the same gene – can have different effects.

For example, two different mutations could affect the primary function of the encoded protein in the same way but have different side effects. One mutation might also cause the protein to interact with a new molecule or protein. Organisms possessing one or the other mutation will thus have similar odds of surviving and reproducing in some environments, but differences in environments where the new interaction is important.

In microorganisms, mutations can lead to drug resistance. If drug-resistant mutations have different side effects, it can be challenging to treat microbial infections, as drug-resistant pathogens are often treated with sequential drug strategies. These strategies rely on mutations that cause resistance to the first drug all having susceptibility to the second drug. But if similar seeming mutations can have diverse side effects, predictions about how they will respond to a second drug are more complicated.

To address this issue, Schmidlin, Apodaca et al. collected a diverse group of nearly a thousand mutant yeast strains that were resistant to a drug called fluconazole. Next, they asked to what extent the fitness – the ability to survive and reproduce – of these mutants responded similarly to environmental change. They used this information to cluster mutations into groups that likely have similar effects at the molecular level, finding at least six such groups with unique trade-offs across environments. For example, some groups resisted only low drug concentrations, and others were unique in that they resisted treatment with two single drugs but not their combination.

These diverse types of fluconazole-resistant yeast lineages highlight the challenges of designing a simple sequential drug treatment that targets all drug-resistant mutants. However, the results also suggest some predictability in how drug-resistant infections can evolve and be treated.

## Introduction

How many different molecular mechanisms can a microbe exploit to adapt to a challenging environment? Answering this question is particularly urgent in the field of drug resistance because infectious populations are adapting to available drugs faster than new drugs are developed (*Centers for Disease Control and Prevention, 2019*, 2019; *Ventola, 2015*). Understanding the mechanistic basis of drug resistance can inform strategies for how to combine existing drugs in a way that prevents the evolution of resistance (*Andersson and Hughes, 2010*; *Melnikov et al., 2020*; *Pinheiro et al., 2021*). For example, one strategy exposes an infectious population to one drug (Drug A) knowing that the mechanism of resistance to Drug A makes cells susceptible to Drug B (*Baym et al., 2016*; *Hall et al., 2009*; *Pál et al., 2015*; *Roemhild et al., 2020*). Problematically, these multi-drug strategies perform best when all mutants that resist Drug A have the same tradeoff in Drug B (*Figure 1A*). If there are multiple different mechanisms to resist Drug A, some of which lack this tradeoff, treatment strategies could fail (*Figure 1B*), and they sometimes do (*Abel zur Wiesch et al., 2014*; *Grier et al., 2003*; *Scarborough et al., 2020*; *Wang et al., 2019*).

Laboratory experiments that have power to search for universal tradeoffs – where all the mutants that perform well in one environment perform poorly in another – often find there are mutants that violate trends or the absence of trends altogether (*Ardell and Kryazhimskiy, 2021*; *Gjini and Wood, 2021*; *Herren and Baym, 2022*; *Hill et al., 2015*; *Kinsler et al., 2020*; *Nichol et al., 2019*). Another way to phrase this observation is to say that adaptive mutations often have collateral effects in environments other than the one in which they originally evolved (*Pál et al., 2015*). But these effects, referred to in some studies as pleiotropic effects, can be unpredictable and context dependent (*Bakerlee et al., 2021*; *Chen et al., 2023*; *Geiler-Samerotte et al., 2020*; *Hinz et al., 2023*; *Jerison et al., 2020*). In simpler terms, some mutants that resist Drug A will suffer a tradeoff in Drug B, but others may suffer a tradeoff in Drug C. To sum, observations from many fields suggest that the mutations that provide a benefit in one environment do not always suffer similar tradeoffs. This begs questions about the extent of diversity among adaptive mutations: does each one suffer a unique set of tradeoffs or can many adaptive mutants be grouped by their common tradeoffs? If there are only a few types of

tradeoff present in a collection of adaptive mutations, multidrug treatments that target tradeoffs may be more feasible.

The goal of the present study is to count how many different types of adaptive mutation, each type being defined by its unique tradeoffs, exist in a population of drug-resistant yeast. This simple counting exercise is surprisingly difficult. One reason why is that the mutations that provide the strongest fitness advantage often dominate evolution. Thus, in the clinic, and in laboratory experiments, the same drug-resistant mutations repeatedly emerge (*Berkow and Lockhart, 2017*; *Ksiezopolska et al., 2021*; *Lupetti et al., 2002*; *Melnikov et al., 2020*), potentially leading to the false impression that the mechanistic basis of resistance to a particular drug, and the associated tradeoffs, are less varied than may be true. This problem is amplified by the limitations of bulk DNA sequencing methods which often miss mutations that are present in less than 10% of a population's cells (*Good et al., 2017*). A similar problem results from strategies to disentangle adaptive from passenger mutations that rely on observing the adaptive ones multiple times in multiple independent replicates (*Martínez and Lang, 2023*). In order to design better multidrug treatment strategies that thwart resistance, or to see if such strategies are feasible, we need methods to survey a more complete set of mutations and mechanisms that can contribute to resistance.

Fortunately, single-cell and single-lineage DNA sequencing technologies are allowing us to more deeply sample genetic diversity in evolving populations of microbes beyond the mutations that dominate evolution (*Schmidt and Efferth, 2016*). Here, we leverage a cutting-edge lineage-tracing method to perform massively replicate evolution experiments in yeast (*Saccharomyces cerevisiae*). This method has been shown to reveal a fuller spectrum of mutations underlying adaptation to a particular environment (*Levy et al., 2015*). One key to its success is that it uses DNA barcodes to track all competing adaptive lineages, not just the ones that ultimately rise to appreciable frequency. Another key feature is that it captures adaptive lineages before they accumulate multiple mutations such that it is easy to pinpoint which mutation is adaptive. We apply this method to investigate mechanisms underlying resistance to a specific antifungal drug: fluconazole (FLU; *Logan et al., 2022*; *Wang et al., 2022*). Although serious fungal infections are most common in immunocompromised individuals, their impact on global health is still striking, resulting in over 1.5 million deaths annually (*Iyer et al., 2022*; *Xie et al., 2014*). By focusing on mechanisms of FLU resistance, we contribute to a growing literature about the tradeoffs that may be leveraged to design multidrug antifungal treatment strategies (*Cowen and Lindquist, 2005*; *Hill et al., 2015*; *Iyer et al., 2022*; *Ksiezopolska et al., 2021*). However, our primary goal is more generic: we seek to explore the utility of a high-throughput evolutionary approach to enumerate classes of drug resistant mutant and their associated tradeoffs.

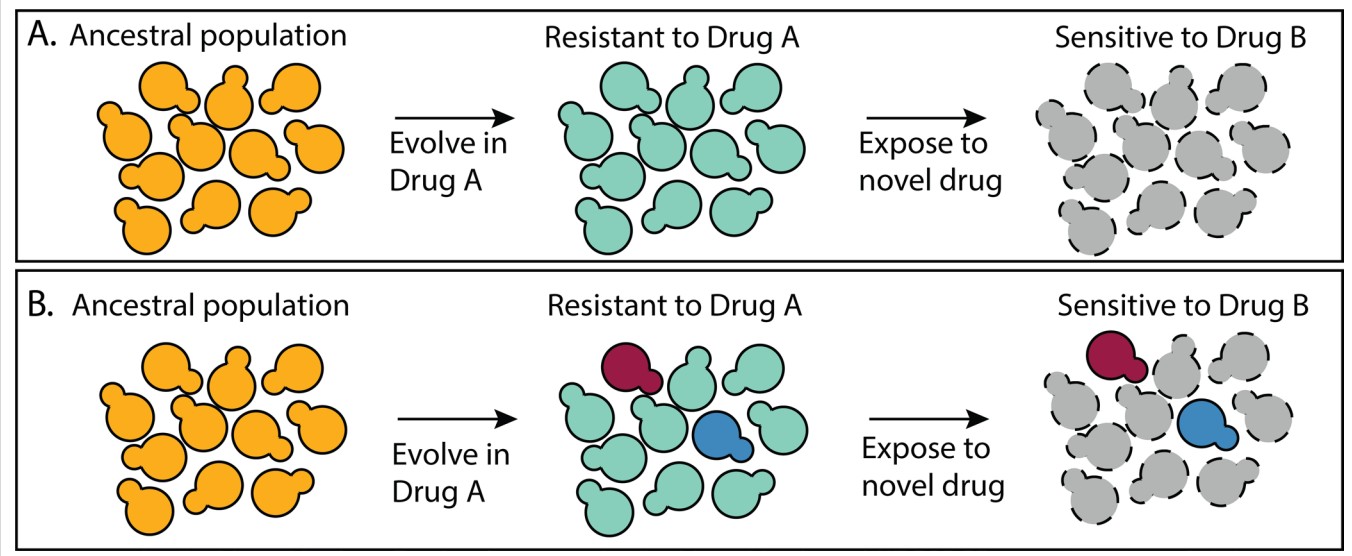

**Figure 1.** A multidrug treatment strategy that relies on all mutants having the same tradeoffs. (**A**) All of the mutants that resist Drug A do so via a similar mechanism such that all are sensitive to Drug B. (**B**) There are multiple different types of mutants that resist Drug A, not all of which are sensitive to Drug B.

To enhance the diversity of drug resistant mutants in our experiment, we performed multiple laboratory evolutions in a range of FLU concentrations and sometimes in combination with a second drug. We did so because previous work has shown that different drug concentrations and combinations select for different azole resistance mechanisms (*Cowen and Lindquist, 2005*; *Hill et al., 2015*). Ultimately, we obtained a large collection of 774 adaptive yeast strains. But how do we know whether we succeeded in isolating diverse types of FLU-resistant mutants? Typical phenotyping methods, for example quantifying expression levels of drug export pumps (*Miyazaki et al., 1998*) or of the drug targets themselves (*Palmer and Kishony, 2014*), are low throughput and require some a priori knowledge of the phenotypes that may be involved in drug resistance. Alternatively, many studies focus on identifying the genetic basis of adaptation in order to glean insights about the underlying mechanisms of resistance (*Cowen et al., 2014*; *Tenaillon et al., 2012*; *Venkataram et al., 2016*). However, genotyping lineages from barcoded pools is technically challenging (*Venkataram et al., 2016*), and further, genotype does not necessarily predict phenotype (*Brettner et al., 2022*; *Eguchi et al., 2019*). For example, previous work using the same barcoded evolution platform used here discovered that the mutations that provide an advantage under glucose-limitation are in genes comprising a canonical glucose-sensing pathway (*Venkataram et al., 2016*). Yet despite this similarity at the genetic level, follow-up work showed that these mutants did not experience the same tradeoffs when exposed to new environments (*Kinsler et al., 2020*; *Li et al., 2018b*).

Instead of trying to identify the phenotypic or even the genetic basis of adaptation, here we strive to enumerate different classes of FLU-resistant mutants. We sort evolved FLU-resistant yeast strains into classes based on whether they share similar tradeoffs across 12 different environments. The intuition here is as follows. If two groups of drug resistant mutants have different fitness tradeoffs, it could mean that they provide resistance through different underlying mechanisms. Alternatively, both could provide drug resistance via the same mechanism, but some mutations might also affect fitness via additional mechanisms (i.e. they might have unique 'side-effects' at the molecular level) resulting in unique fitness tradeoffs in some environments. Previous work is consistent with the idea that mutants with different fitness tradeoffs affect fitness through different underlying mechanisms (*Li et al., 2019*; *Pinheiro et al., 2021*; *Rodrigues et al., 2016*). Our work can be seen as part of a growing push to flip the problem of mechanism on its head (*Kinsler et al., 2020*; *Li et al., 2019*; *Petti et al., 2023*). Instead of using a mechanistic understanding to predict a microbe's fitness, here we use how fitness varies across environments to distinguish mutants that likely affect fitness via different mechanisms. This inverted approach to investigating the mechanisms by which mutations affect fitness has broad applications; it could be used to characterize dominant negative mutations (*Flynn et al., 2024*; *Padhy et al., 2023*), mutations with collateral fitness effects (*Mehlhoff et al., 2020*; *Mehlhoff and Ostermeier, 2023*), and in other high-throughput mutational scanning studies (*Flynn et al., 2020*; *Fowler and Fields, 2014*; *Hinz et al., 2023*; *Starr et al., 2017*).

The key requirement to being able to implement this approach is having a large collection of barcoded mutants and the ability to re-measure their fitness, relative to a reference strain, in multiple environments, such as the 12 different combinations and concentrations of drugs surveyed here. Across our collection of 774 adaptive yeast lineages, we discovered at least 6 distinct groups with characteristic fitness tradeoffs across these 12 environments. For example, we find some drug resistant mutants are generally advantageous, while others are advantageous only in specific environments. And we find some mutants that resist single drugs also resist combinations of those drugs, while others do not. By grouping mutants with similar tradeoffs, we reduce the number of unique drug-resistant mutants from more than can be easily phenotyped (774) to a manageable panel of six types for investigating the molecular mechanisms by which mutations impact fitness.

With regard to multidrug regimens that exploit tradeoffs (*Figure 1*), our finding of multiple mutant classes with different tradeoffs suggests this may not be straightforward. The outlook is further complicated by our finding that some classes of FLU-resistant mutant primarily emerge from evolution experiments that did not contain FLU. This, as well as limits on our power to observe mutants with strong tradeoffs, suggest there may be additional types of FLU-resistant mutant beyond those we sampled. These observations suggest multidrug strategies that assume resistant mutants suffer consistent or common tradeoffs will often fail.

On the other hand, nuanced strategies to forestall resistance that allow for multiple mutant types are emerging (*Gjini and Wood, 2021*; *Maltas and Wood, 2019*; *Wang et al., 2023*). For example,

one idea is to apply a drug regimen that enriches for mutants that suffer a particular tradeoff before exploiting that tradeoff (*Iram et al., 2021*). Another idea is to perform single-cell sequencing on infectious populations to discover which classes of mutants are present (*Forsyth et al., 2021*; *Nagasawa et al., 2021*) and design treatments specific to those (*Aissa et al., 2021*; *Hsieh et al., 2022*; *Maltas and Wood, 2019*). Our findings support that such ideas may be feasible by demonstrating that there are not as many unique fitness tradeoffs as there are mutations.

More generally, our work – showing that 774 mutants fall into a much smaller number of groups – contributes to growing literature suggesting that the phenotypic basis of adaptation is not as diverse as the genetic basis (*Iwasawa et al., 2022*; *Kinsler et al., 2020*; *Petti et al., 2023*). This winnowing of diversity is important: it may mean that evolutionary processes, for example, whether an infectious population will adapt to resist a drug, are sometimes predictable (*King et al., 2022*; *Kinsler et al., 2020*; *Lässig et al., 2017*; *Rodrigues et al., 2016*; *Wortel et al., 2023*; *Yoon et al., 2021*).

## Results
### Barcoded evolution experiments uncover hundreds of yeast lineages with adaptive mutations

In order to create a sizable collection of drug-resistant mutants, we performed high-replicate evolution experiments utilizing barcoded yeast (*S. cerevisiae*; *Boyer et al., 2021*; *Levy et al., 2015*; *Li et al., 2018b*). This barcoding system allows evolving hundreds of thousands of genetically identical yeast lineages together in a single flask. Each lineage is tagged with a unique DNA barcode, which is a 26 base pair sequence of DNA located within an artificial intron. Lineages with unique barcodes can be thought of as independent replicates of an evolution experiment. This high-replicate system has the potential to generate many different yeast lineages that differ by single adaptive mutations (*Kinsler et al., 2020*; *Venkataram et al., 2016*).

We performed a total of 12 barcoded evolution experiments, each of which started from the same pool of approximately 300,000 barcoded yeast lineages (*Figure 2A, B*; *Figure 2—figure supplement 1*). These evolutions survey how yeast cells adapt to different concentrations and combinations of two drugs: fluconazole (FLU) and radicicol (RAD). FLU is a first line of defense against yeast infections, but over the past two decades diverse resistant mutations have been identified (*Bongomin et al., 2017*; *Osset-Trénor et al., 2023*; *Rybak et al., 2022*). Some earlier work suggested that FLU-resistant mutants are sensitive to the second drug we study, radicicol (RAD; *Cowen et al., 2009*; *Cowen and Lindquist, 2005*), and more generally that RAD can prevent the emergence of drug resistance in other systems (*Whitesell et al., 2014*). However, there are some mutants that are cross-resistant to both FLU and RAD (*Hill et al., 2015*), and the prominent mechanism of resistance can differ with the intensity of selection and drug concentration (*Cowen and Lindquist, 2005*; *Yang et al., 2023*). We thus chose to evolve yeast to resist different concentrations and combinations of FLU and RAD to generate a diverse pool of adaptive mutations comprising different mechanisms of drug resistance.

We evolved yeast to resist three different concentrations of either FLU or RAD for a total of six single-drug conditions (*Table 1*). We also studied four conditions containing combinations of both drugs, as well as two control conditions, for a total of 12 evolution experiments (*Table 1*). We chose to study subclinical drug concentrations with the hope that no drug treatment would be strong enough to reduce the population of yeast cells to only a handful of unique barcodes (*Figure 2—figure supplement 2*). We needed to maintain barcode diversity in order to evolve a large number of unique lineages that each accumulate different mutations.

With the goal of collecting adaptive lineages from each evolution experiment, we took samples from each one after 3–6 growth/transfer cycles (*Figure 2—figure supplement 1*). This represents roughly 24–48 generations of growth assuming 8 generations per growth/transfer cycle (*Levy et al., 2015*). We sampled early because previous work using this barcoded evolution system demonstrated that the diversity of adaptive lineages peaks after just a few dozen generations (*Levy et al., 2015*; *Venkataram et al., 2016*). This happens because the barcoding process is slightly mutagenic, thus there is less need to wait for DNA replication errors to introduce mutations (*Levy et al., 2015*; *Venkataram et al., 2016*). We sampled about 2000 cells from each evolution experiment except those three containing high concentrations of FLU from which we sampled only 1000 cells for a total of ~21,000 isolates (2000 cells x 9 conditions +1000 cells x 3 conditions) (*Figure 2C*).

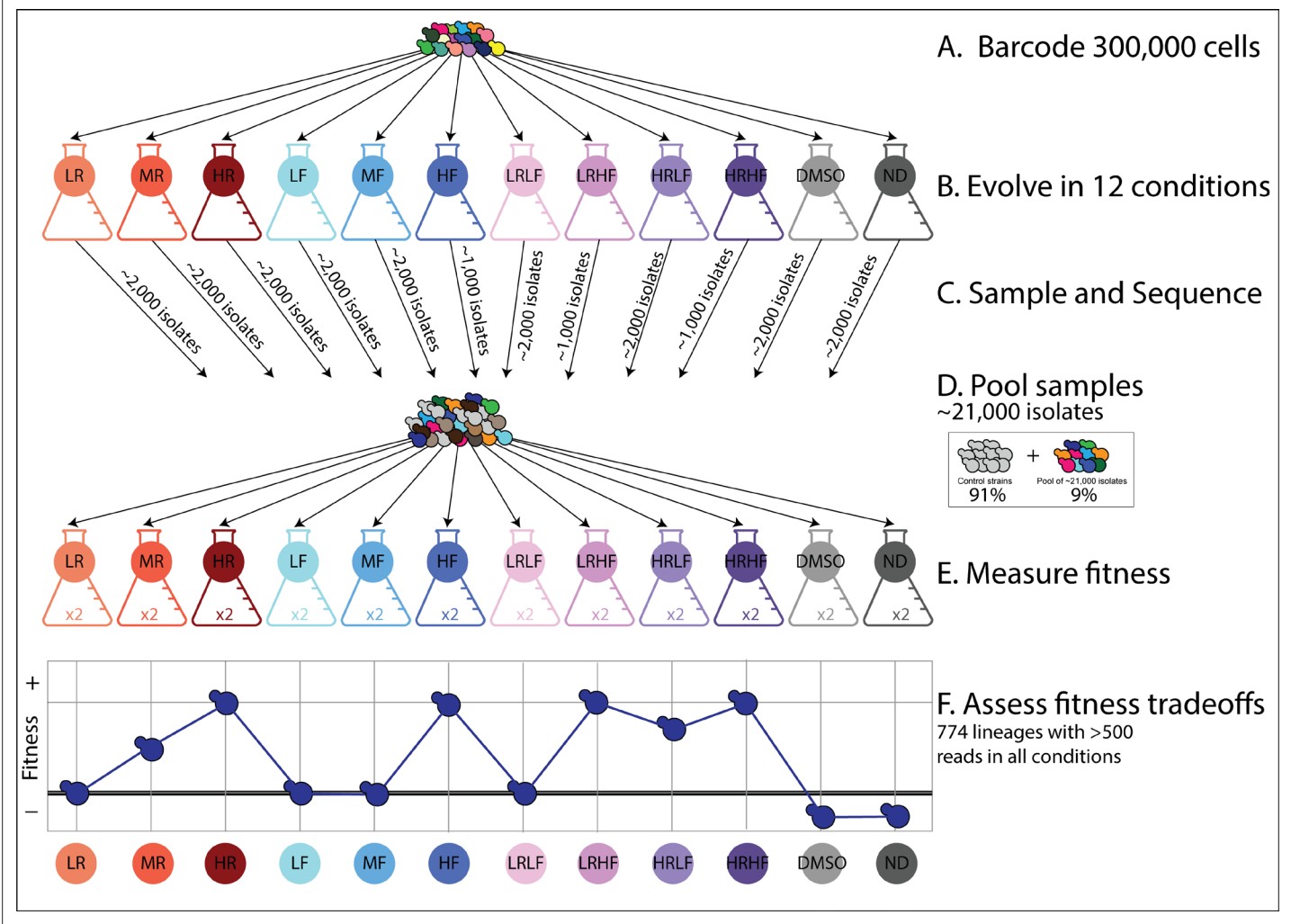

**Figure 2.** An overview of the experimental design. (**A**) Yeast cells were barcoded to create 300,000 lineages. (**B**) These lineages were evolved in 12 different conditions (**Table 1**). (**C**) A small sample of evolved isolates were taken from each evolution experiment and their barcodes were sequenced. These ~21,000 isolates do not represent as many unique, adaptive lineages because many either have the same barcode or do not possess adaptive mutations. (**D**) These samples of evolved isolates were all pooled together with control strains representing the ancestral genotype. (**E**) Barcoded fitness competition experiments were then performed on this pool in each of the 12 evolution conditions. Fitness was measured by tracking changes in each barcode's frequency over time relative to control strains. Two replicates per condition were performed. (**F**) The overall goal is to investigate fitness tradeoffs for hundreds of adaptive lineages. For example, the adaptive lineage depicted in dark blue has higher fitness than the ancestor in some environments (HR, HF) but lower fitness in others (DMSO, ND). We were able to investigate fitness tradeoffs for 774 adaptive lineages. We excluded lineages when we did not observe their associated barcode at least 500 times in all 12 environments. In other words, we only included lineages for which we obtained high-quality fitness estimates in all 12 environments.

The online version of this article includes the following figure supplement(s) for figure 2:

**Figure supplement 1.** Twelve barcoded evolution experiments track ~300,000 lineages as they adapt to different drug concentrations and combinations.

**Figure supplement 2.** Chosen drug concentrations do not dramatically reduce yeast's maximum growth rate.

**Figure supplement 3.** Twenty-four fitness competitions track evolved lineages as their barcodes change frequency.

**Figure supplement 4.** Fitness measurements are reproducible between replicates and closely related conditions.

Next, we measured the fitness of each isolate in each of the 12 evolution environments to quantify fitness tradeoffs (e.g. whether mutants that perform well in one environment perform worse in another). This process also indirectly screens isolates for adaptive mutations by comparing the fitness of each evolved isolate to the ancestor of the evolution experiments (**Venkataram et al., 2016**). To do so, we pooled these 21,000 isolates and used this pool to initiate fitness competition experiments

**Table 1.** A list of the environments included in this study and the symbol used to represent them in 242 subsequent figures.

| Evolution Condition | Abbreviation | Symbol |
|---|---|---|
| Fluconazole | Low Flu | Low Flu |
| Fluconazole | Med Flu | Med Flu |
| Fluconazole | High Flu | High Flu |
| Radicicol | Low Rad | Low Rad |
| Radicicol | Med Rad | Med Rad |
| Radicicol | High Rad | High Rad |
| Radicicol Fluconazole | LRLF | LRLF |
| Radicicol Fluconazole | LRHF | LRHF |
| Radicicol Fluconazole | HRLF | HRLF |
| Radicicol Fluconazole | HRHF | HRHF |
| DMSO | DMSO | DMSO |
| No Drug | No Drug | No Drug |

(*Figure 2D*). We competed the pool against control strains, that is strains of the ancestral genotype that do not possess adaptive mutations (*Kinsler et al., 2020*; *Venkataram et al., 2016*). We performed 24 such competitive fitness experiments, 2 per each of the original 12 evolution conditions (*Figure 2E*). In each experiment, we emulated the growth and transfer conditions of the original evolution experiments as precisely as possible, tracking how barcode frequencies changed over five growth/transfer cycles (~40 generations). We used the log-linear slope of this change, relative to the average slope for the control strains, to quantify relative fitness.

We found that many barcoded lineages have higher fitness than the control strains in some conditions, presumably because they possess adaptive mutations that improve their fitness in some conditions (*Figure 2—figure supplement 3*). In fact, some of these adaptive lineages outcompeted the other lineages so quickly that it posed a challenge. Barcodes pertaining to outcompeted lineages were often not present at high enough coverage to track their fitness. We applied a conservative filter, preserving only 774 lineages with barcodes that were observed >500 times in at least one replicate experiment per each of the 12 conditions. The reason we required fitness measurements in all 12 conditions is that our goal is to examine each lineage's fitness tradeoffs (*Figure 2F*) in order to see if different lineages have different tradeoffs. In order to compare apples to apples, we need to measure each lineage's fitness in the same set of environments.

The 774 lineages we focus on are biased towards those that are reproducibly adaptive in multiple environments we study. This is because lineages that have low fitness in a particular environment are rarely observed >500 times in that environment (*Figure 2—figure supplement 4*). By requiring lineages to have high-coverage fitness measurements in all 12 conditions, we exclude adaptive mutants that have severe tradeoffs in one or more environments, consequently blinding ourselves to mutants that act via unique underlying mechanisms. Despite this bias, we will go on to demonstrate that there are different types of mutants with characteristically different fitness tradeoffs present among the 774 lineages that remain.

To provide additional evidence that these 774 barcoded yeast lineages indeed possess adaptive mutations, we performed whole genome sequencing on a subset of 53 lineages. Because we sampled these lineages after only a few dozen generations of evolution, each lineage differs from the ancestor by one or just a few mutations, making it easy to pinpoint the genetic basis of adaptation. Doing so revealed mutations that have previously been shown to be adaptive in our evolution conditions (*Supplementary file 1*). For example, we sequenced many FLU-resistant yeast lineages finding 35 with unique single nucleotide mutations in either PDR1 or PDR3, and a few with mutations in SUR1 or UPC2, genes which have all been shown to contribute to FLU resistance in previous work (*Flowers et al., 2012*; *Tanaka and Tani, 2018*; *Uemura and Moriguchi, 2022*; *Vasicek et al., 2014*; *Vu and Moye-Rowley, 2022*). Similarly, lineages that have very high fitness in RAD were found to possess single nucleotide mutations in genes associated with RAD resistance, such as HDA1 (*Robbins et al., 2012*) and HSC82, which is the target of RAD (*Roe et al., 1999*). We also observed several lineages with similar mutations to those observed in other studies using this barcoded evolution regime, including mutations to IRA1, IRA2, and GPB2 (*Kinsler et al., 2020*; *Venkataram et al., 2016*). Previous barcoded evolutions also observed that increases in ploidy were adaptive, with 43% to 60% of cells becoming diploid during the course of evolution (*Venkataram et al., 2016*). However, ploidy changes contributed less to adaptation in our experiment, with at most 9.4% of cells becoming diploid by the time point when we sampled, but often less than 2% (*Supplementary file 2*).

In sum, we have created a diverse pool of 774 barcoded yeast lineages, most of which have a fitness advantage in at least one of the conditions we study and are likely to possess a unique adaptive mutation. The question we address for the rest of this study is to what extent these hundreds of mutant lineages differ from one another in terms of their fitness tradeoffs and the mechanism/s underlying their fitness advantages.

## A unique mechanisms of FLU resistance emerges among mutants isolated in RAD evolutions

The majority of the 774 adaptive lineages that we study have higher fitness than the ancestral strains in not one, but often in several drug conditions. This suggests that pleiotropy, and in particular cross-resistance, is prevalent among the lineages we study. But not all lineages show the same patterns of cross-resistance (*Figure 3*). For example, the 100 most fit lineages in our highest concentration of fluconazole are also beneficial in our highest concentration of radicicol (*Figure 3A*; leftmost two boxplots). As expected, these 100 lineages also have high fitness in conditions where high concentrations of FLU and RAD are combined (*Figure 3A*; third boxplot). And these 100 most-fit lineages in FLU lose their fitness advantage in conditions where no drug is present (*Figure 3A*; rightmost boxplot).

Given their high fitness in conditions containing FLU, it seems likely that these 100 mutants originated from evolution experiments containing FLU. We can trace every lineage back to the evolution experiment/s it originated from because we sequenced the lineages we sampled from each evolution experiment before pooling all 21,000 isolates (*Figure 2C*). As we expected, these 100 best performing lineages in high FLU largely originate from evolution experiments containing FLU (*Figure 3B*). Given that these lineages have no fitness advantage in conditions containing no drug, it is also unsurprising that they are underrepresented in evolution experiments lacking RAD and FLU (*Figure 3B*).

It might be tempting to generalize that most mutations that provide drug resistance are not beneficial in environments without drugs. Afterall, we show this is true for 100 independent lineages (*Figure 3A*). Further, many previous studies find a similar pattern, whereby drug resistant mutants often do not have high fitness in the absence of drug (*Allen et al., 2019*; *Andersson and Hughes, 2010*; *Basra et al., 2018*; *Melnikov et al., 2020*), such that treatment strategies have emerged that cycle patients between drug and no drug states, albeit with mixed success (*Algazi et al., 2020*;

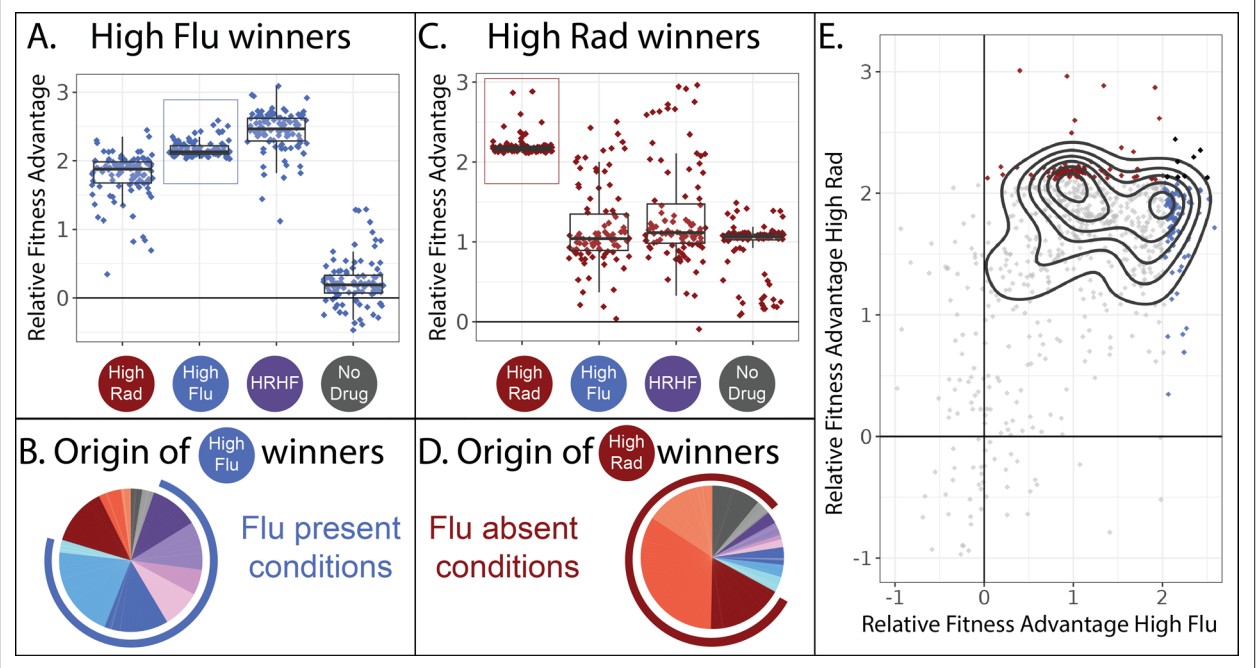

**Figure 3.** Two different classes of FLU-resistant mutants with unique tradeoffs. (**A**) This panel describes the 100 mutant lineages with the highest fitness relative to the control strains in the high FLU environment (8 µg/ml FLU). The vertical axis depicts the fitnesses (log-linear slopes relative to control strains) for these 100 strains in four selected environments, including the high FLU environment (boxed). Boxplots summarize the distribution across all 100 lineages for each environment, displaying the median (center line), interquartile range (IQR) (upper and lower hinges), and highest value within 1.5 × IQR (whiskers). (**B**) The 100 lineages with highest fitness in high FLU were most often sampled from evolution experiments in which FLU was present. In this pie-chart, colors correspond to the evolution conditions listed in **Table 1** and the blue outer ring highlights evolution conditions that contain FLU. The size of each slice of pie represents the relative frequency with which these 100 lineages were found in each evolution experiment. (**C**) Similar to panel A, this panel describes the 100 mutant lineages with the highest fitness relative to the control strains in the high RAD environment (20 µM Rad). (**D**) The 100 lineages with highest fitness in high RAD were most often sampled from evolution experiments that did not contain FLU. (**E**) A pairwise correlation plot showing that all 774 mutants, not just the two groups of 100 depicted in panels A and C, to some extent fall into two groups defined by their fitness in high FLU and high RAD. The contours (black curves) were generated using kernel density estimation with bins = 7. These contours describe the density of the underlying data, which is concentrated into two clusters defined by the two smallest black circles. The 100 mutants with highest fitness in high FLU are blue, highest fitness in high RAD are red, and the seven that overlap between the two aforementioned categories are black.

The online version of this article includes the following figure supplement(s) for figure 3:

**Figure supplement 1.** The two types of adaptive mutants depicted in **Figure 3** sort into different clusters on the UMAP.

**Baker et al., 2018**; **Raymond, 2019**; **Wang et al., 2019**). However, this type of generalization is not supported by our data. We find that drug resistance can sometimes come with an advantage, rather than a cost, in the absence of a drug (**Figure 3C**). The top 100 most fit mutants in our highest concentration of RAD provide a fitness advantage in high RAD, high FLU, as well as in environments with no drug (**Figure 3C**). These observations suggest that there are at least two different mechanisms by which to resist FLU that result in different tradeoffs in other environments (**Figure 3A** vs. **3** C).

Intriguingly, these FLU-resistant lineages that maintain their fitness advantage in the absence of drug (**Figure 3C**) mainly originate from evolution experiments performed in conditions lacking FLU (**Figure 3D**). This highlights how the potential mechanisms by which a microbe can resist a drug may be more varied than is often believed. Typically, one does not search for FLU-resistant mutants by evolving yeast to resist RAD. Thus typical studies might miss this unique class of FLU-resistant mutants.

In sum, there appear to be at least two different types of mutants present among our collection of 774 adaptive yeast lineages. One group has almost equally high fitness in RAD and FLU but has no fitness advantage over the ancestral strain in conditions without either drug (**Figure 3A**). Another group is defined by very high fitness in RAD, moderately high fitness in FLU and moderately high fitness in conditions without either drug (**Figure 3C**). When comparing fitness in RAD vs. FLU across all 774 lineages, not only the top 100 best performing in each drug, we see some evidence that they

largely fall into the two main categories highlighted in *Figure 3A and C* (*Figure 3E*). Thus, it might be tempting to conclude that there are two different types of FLU-resistant mutant in our dataset. However, sorting mutants into groups using a pairwise correlation plot (*Figure 3E*) excludes data from 10 of our 12 environments.

## A strategy to differentiate classes of drug-resistant mutants with different tradeoffs

The observation of two distinct types of adaptive mutants (*Figure 3*) made us wonder whether there were additional unique types of FLU-resistant mutants with their own characteristic tradeoffs. This is difficult to tell by using pairwise correlation like that in *Figure 3E* because we are not studying pairs of conditions, as is somewhat common when looking for tradeoffs to leverage in multidrug therapies (*Ardell and Kryazhimskiy, 2021*; *Larkins-Ford et al., 2022*; *Melnikov et al., 2020*; *Scarborough et al., 2020*). Instead, we have collected fitness data from 12 conditions to yield a more comprehensive set of gene-by-environment interactions for each mutant. This type of data, describing how a particular genotype responds to environmental change, is sometimes called a 'reaction norm' and can inform quantitative genetic models of how selection operates in fluctuating environments (*Gomulkiewicz and Kirkpatrick, 1992*; *Ogbunugafor, 2022*) and how much pleiotropy exists in nature (*Yadav et al., 2015*). More recent studies refer to the changing performance of a genotype across environments as a 'fitness profile' or in aggregate, a 'fitness seascape', and suggest these type of dynamic measurements are the key to designing effective multi-drug treatments (*King et al., 2022*) and to predicting evolution (*Cairns et al., 2022*; *Chen et al., 2023*; *Iwasawa et al., 2022*; *Kinsler et al., 2020*; *Lässig et al., 2017*). And when the environments studied represent different drugs, these types of data are often referred to as 'collateral sensitivity profiles' a term chosen to convey how resistance to one drug can have 'collateral' effects on performance in other drugs (*Gjini and Wood, 2021*; *Maltas and Wood, 2019*; *Pál et al., 2015*). Despite the wide interest in this type of fitness data, it is technically challenging to generate, thus many previous studies of fitness profiles focus on a much smaller number of isolates (*Imamovic et al., 2018*; *Maltas and Wood, 2019*; *Nichol et al., 2019*), sometimes with variation restricted to a single gene (*King et al., 2022*; *Mira et al., 2015*), or evolved in response to a single selection pressure (*Kinsler et al., 2020*; *Li et al., 2018b*). Here, we have generated fitness profiles for a large and diverse group of drug-resistant strains using the power of DNA barcodes. Now we seek to understand whether these mutants fall into distinct classes that each have characteristic fitness profiles (i.e. characteristic tradeoffs, characteristic reaction norms, or characteristic gene-by-environment interactions).

To address this question, we start by performing dimensional reduction, clustering mutants with fitness profiles that have a similar shape. It is in theory possible for all mutants to have similar profiles, perhaps implying they all affect fitness through similar underlying mechanisms (*Figure 4A*). However, the disparity reported in *Figure 3* suggests otherwise. It is also possible that every mutant will have a different profile. This could happen if each mutant affects different molecular-level phenotypes that underlie its drug resistance (*Figure 4B*). But previous work suggests that the phenotypic basis of adaptation is less diverse than the genotypic basis (*Brettner et al., 2024*; *Iwasawa et al., 2022*; *Kinsler et al., 2020*). A final possibility, somewhere in between the first two, is that there exist multiple classes of drug-resistant mutants each with characteristic tradeoffs (*Figure 4C*). This might imply that each class of mutants provides drug resistance via a different molecular mechanism, or a different set of mechanisms. Overall, our endeavor to enumerate how many distinct fitness profiles are present across these 774 mutants (*Figure 4A – C*) informs general questions about the extent of pleiotropy in the genotype-phenotype-fitness map (*Bakerlee et al., 2021*; *Boyle et al., 2017*; *Chen et al., 2023*; *Geiler-Samerotte et al., 2020*), the extent to which fitness tradeoffs are universal (*Andersson and Hughes, 2010*; *Herren and Baym, 2022*; *Li et al., 2019*), and relatedly, the extent to which evolution is predictable (*Iram et al., 2021*; *King et al., 2022*; *Kinsler et al., 2020*; *Lässig et al., 2017*; *Petti et al., 2023*).

To see whether there are distinct fitness profiles present among our drug-resistant yeast lineages, we applied uniform manifold approximation and projection (UMAP) to fitness measurements for 774 yeast strains across all 12 environments. This method places mutants with similar fitness profiles near each other in two-dimensional space. As might be expected, it largely places mutants in each of the two categories described in *Figure 3* far apart, with drug-resistant mutants that lose their benefit in

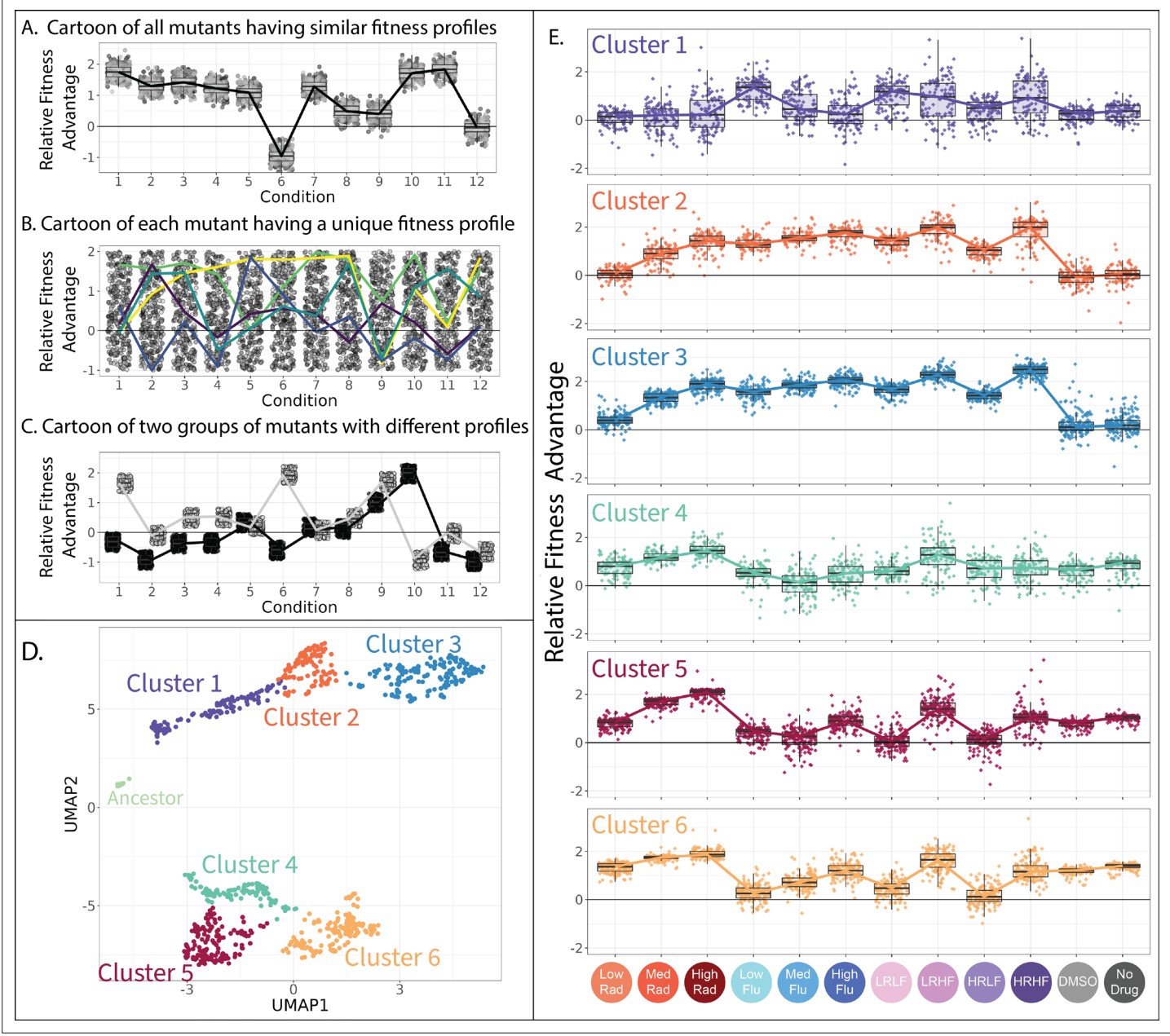

**Figure 4.** Clustering evolved lineages with similar fitness profiles. (**A–C**) Simulated data showing potential fitness profiles when (**A**) all mutants have similar responses to environmental change and thus a similar fitness profile, (**B**) every mutant has a different profile (five unique profiles are highlighted in color), or (**C**) every mutant has one of a small number of unique profiles (two unique profiles are depicted). (**D**) Every point in this plot represents one of the barcoded lineages colored by cluster; clusters were identified using a gaussian mixture model. The 774 adaptive lineages cluster into 6 groups based on variation in their fitness profiles; the control lineages cluster separately into the leftmost cluster in light green. (**E**) The fitness profiles of each cluster of adaptive lineages. Boxplots summarize the distribution across all lineages within each cluster in each environment, displaying the median (center line), interquartile range (IQR) (upper and lower hinges), and highest value within 1.5×IQR (whiskers).

The online version of this article includes the following figure supplement(s) for figure 4:

**Figure supplement 1.** Bayesian information criteria (BIC) scores suggest the 774 mutants cluster into between 6 and 13 groups.

**Figure supplement 2.** UMAP structure is robust.

**Figure supplement 3.** Clusters are robust to a hierarchical clustering method.

**Figure supplement 4.** Clusters are robust to principal component analysis.

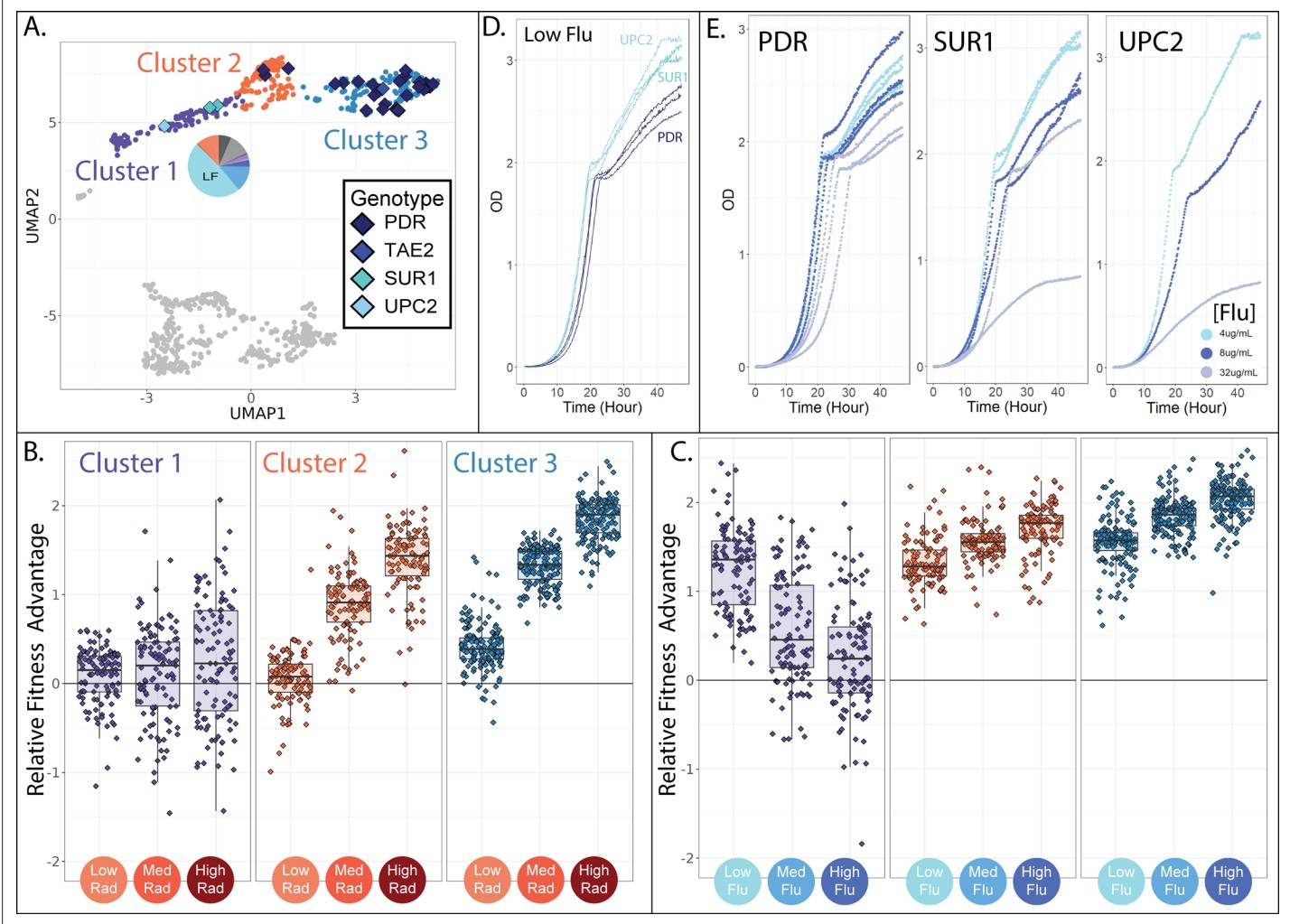

**Figure 5.** Evolved lineages comprising cluster 1 have different genotypes and phenotypes from neighboring clusters. (**A**) The three clusters on the top half of the UMAP differ in their genetic targets of adaptation with cluster 1 being unique in that it does not contain mutations to PDR1 or PDR3. Cluster 1 is also unique in that it contains lineages that predominantly originated from the low fluconazole evolution condition; the pie chart depicts the fraction of lineages originating from each of the 12 evolution environments with colors corresponding to *Table 1*. (**B**) Evolved lineages comprising cluster 1 do not have consistent fitness advantages in conditions containing RAD, while lineages comprising clusters 2 and 3 are uniformly adaptive in medium and high RAD. Boxplots summarize the distribution across all lineages within each cluster in each environment, displaying the median (center line), interquartile range (IQR) (upper and lower hinges), and highest value within 1.5×IQR (whiskers). (**C**) Lineages comprising cluster 1 are most fit in low concentrations of FLU, and this advantage dwindles as the FLU concentration increases. Lineages comprising clusters 2 and 3 show the opposite trend. (**D**) In low FLU (4 µg/ml), Cluster 1 lineages (UPC2 and SUR1) grow faster and achieve higher density than lineages from cluster 3 (PDR). This is consistent with bar-seq measurements demonstrating that cluster 1 mutants have the highest fitness in low FLU. (**D**) Cluster 1 lineages are sensitive to increasing FLU concentrations (SUR1 and UPC2). This is apparent in that the dark blue (8 µg/ml flu) and grey (32 µg/ml flu) growth curves rise more slowly and reach lower density than the light blue curves (4 µg/ml flu). But this is not the case for the PDR mutants. These observations are consistent with the bar-seq fitness data (*Figure 4E*).

the absence of drug in the top half of the graph, and those that maintain their benefit in the bottom half (*Figure 4D* and *Figure 3—figure supplement 1*).

Beyond the obvious divide between the top and bottom clusters of mutants on the UMAP, we used a gaussian mixture model (GMM; *Fraley and Raftery, 2003*) to identify clusters. A common problem in this type of analysis is the risk of dividing the data into clusters based on variation that represents measurement noise rather than reproducible differences between mutants (*Mirkin, 2011*; *Zhao et al., 2008*). One way we avoided this by using a GMM quality control metric (BIC score) to establish how splitting out additional clusters affected model performance (*Figure 4—figure supplement 1*). Another factor we considered were follow-up genotyping and phenotyping studies that

demonstrate biologically meaningful differences between mutants in different clusters (*Figure 5*, *Figure 6*, *Figure 7*, *Figure 8*). Using this information, we identified seven clusters of distinct mutants, including one pertaining to the control strains, and six others pertaining to presumed different classes of adaptive mutant (*Figure 4D*). It is possible that there exist additional clusters, beyond those we are able to tease apart in this study.

Preliminarily, we investigated whether the clusters we identified capture reproducible differences between mutants, rather than measurement noise, by reducing the amount of noise in our data and asking if the same clusters are still present. To do so, we reduced our collection of adaptive lineages from 774 to 617 by requiring 5000 rather than 500 reads per lineage per experiment in order to infer fitness. This procedure reduced noise; the Pearson correlation across replicate experiments improved from 0.756% to 0.813%. Despite this reduction in variation, these 617 lineages cluster into the same six groups (plus a seventh pertaining to the control strains) as do the original 774 (*Figure 4—figure supplement 2*). The groupings are also preserved when we perform alternate methods for dimensionality reduction (*Figure 4—figure supplement 3* and *Figure 4—figure supplement 4*).

Each of the six clusters of adaptive mutants has a characteristic fitness profile (*Figure 4E*). In any given environment, the fitnesses of the mutants within each cluster are often very similar to one another and often significantly different from other clusters (*Figure 4E*). Our follow-up investigations (*Figures 5–8*), including whole genome sequencing, growth rate measurements, and tracing the evolutionary origins of the mutants in each cluster, provide additional evidence that the adaptive mutants in each cluster have characteristically different properties.

## A group of mutants with distinct genotypes are primarily resistant to low concentrations of FLU

The upper three clusters of mutants on the UMAP (*Figure 4D*) are all similar in that they have elevated fitness in at least one FLU-containing environment but ancestor-like fitness in the absence of drug (*Figure 4E*; upper three profiles). Despite these similarities, there are major differences between these three groups of mutant lineages, both at the level of genotype and fitness profile (*Figure 5*). For example, in cluster 1 (depicted in purple in *Figures 4 and 5*), the three sequenced lineages have single nucleotide mutations to either SUR1 or UPC2 (*Figure 5A*). But in clusters 2 and 3 (depicted in blue and orange in *Figures 4 and 5*), 35/36 sequenced lineages have unique single nucleotide mutations to one of two genes associated with 'Pleiotropic Drug Resistance' (PDR1 or PDR3).

PDR1 and PDR3 are transcription factors that are well known to contribute to fluconazole resistance through increased transcription of a drug pump (PDR5) that removes FLU from cells (*Fardeau et al., 2007*; *Osset-Trénor et al., 2023*). However, SUR1 and UPC2 are less commonly mentioned in literature pertaining to FLU resistance, and have different functions within the cell as compared to PDR1 and PDR3 (*Hill et al., 2015*; *Kapitzky et al., 2010*). SUR1 converts inositol phosphorylceramide to mannosylinositol phosphorylceramide, which is a component of the plasma membrane (*Uemura and Moriguchi, 2022*). Similarly, UPC2 is a transcription factor with a key role in activating the ergosterol biosynthesis genes, which contribute to membrane formation (*Tan et al., 2022*; *Rine, 2001*). The presence of adaptive mutations in genes involved in membrane synthesis is consistent with fluconazole's disruptive effect on membranes (*Sorgo et al., 2011*).

Interestingly, the lineages with mutations to UPC2 and SUR1, and the unsequenced lineages in the same cluster, do not consistently have cross-resistance in RAD (*Figure 5B*; cluster 1). Oppositely, lineages with mutations to PDR1 or PDR3, and the unsequenced lineages in the same clusters, are uniformly cross-resistant to RAD (*Figure 5B*; clusters 2 and 3). Perhaps, this cross-resistance is reflective of the fact that the drug efflux pump that PDR1/3 regulates (PDR5) can transport a wide range of drugs and molecules out of yeast cells (*Harris et al., 2021*; *Kolaczkowski et al., 1996*). Overall, the targets of adaptation in cluster 1 have disparate functions within the cell as compared to the targets of adaptation in clusters 2 and 3. This may suggest that the mutants in cluster 1 confer FLU resistance via a different mechanism than clusters 2 and 3.

The lineages in cluster 1 have additional important differences from clusters 2 and 3. The lineages in cluster 1 perform best in the lowest concentration of FLU and have decreasing fitness as the concentration of FLU rises (*Figure 5C*). In fact, about 15% of these mutant lineages perform worse than their ancestor in the highest concentration of FLU, suggesting the very mutations that provide resistance to low FLU are costly in higher concentrations of the same drug. The mutants in clusters 2 and 3 show

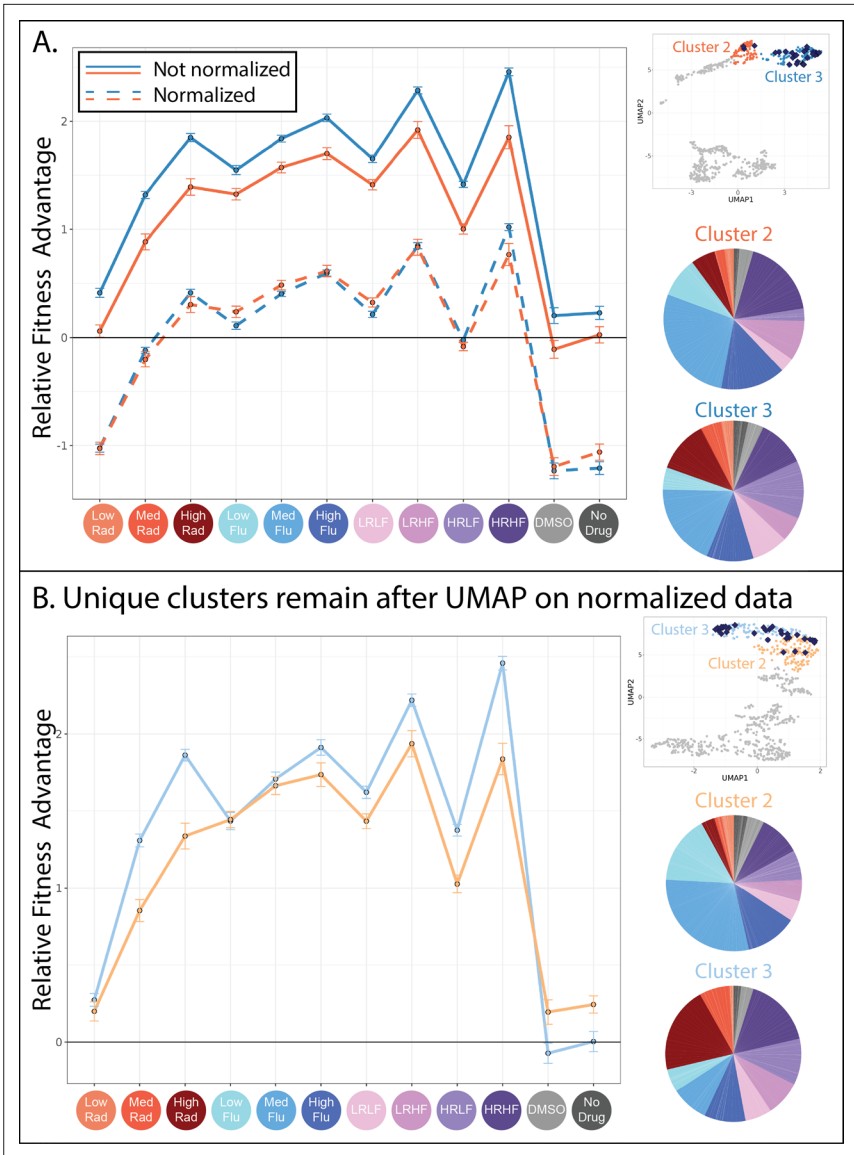

**Figure 6.** Evolved lineages in clusters 2 and 3 have characteristic differences despite similarities at the genetic level. (**A**) This panel shows the similarities between clusters 2 and 3. The upper right inset displays the same UMAP from *Figure 4D* with only clusters 2 and 3 highlighted and with lineages possessing mutations to the PDR genes depicted as blue diamonds. The line plot displays the same fitness profiles for clusters 2 and 3 as *Figure 4E*, plotting the average fitness for each cluster in each environment and a 95% confidence interval. Dotted lines represent the same data, normalized such that every lineage has an average fitness of 0 across all environments. These line plots show that the fitness profiles for clusters 2 and 3 have a very similar shape. Pie charts display the relative frequency with which lineages in clusters 2 and 3 were sampled from each of the 12 evolution conditions, colors match those in the horizontal axis of the line plot and *Table 1*. (**B**) This panel shows the differences between the new clusters 2 and 3 created after all fitness profiles were normalized to eliminate magnitude differences. The upper right inset displays a new UMAP (also see *Figure 6—figure supplement 1*) that summarizes variation in fitness profiles after each profile was normalized by setting its average fitness to 0. The line plot displays the fitness profiles for the new clusters 2 and 3, which look different from those in panel A because 37% of mutants in the original clusters 2 and 3 switched identity from 2 to 3 or vice versa. The new clusters 2 and 3 are depicted in slightly different shades of blue and orange to reflect that these are not the same groupings as those depicted in *Figure 4*. Pie charts display the relative frequency with which lineages in new clusters 2 and 3 were sampled from each of the 12 evolution conditions, colors match those in the horizontal axis of the line plot and *Table 1*.

The online version of this article includes the following figure supplement(s) for figure 6:

**Figure supplement 1.** UMAP on data that were normalized to account for magnitude differences (row means set to 0).

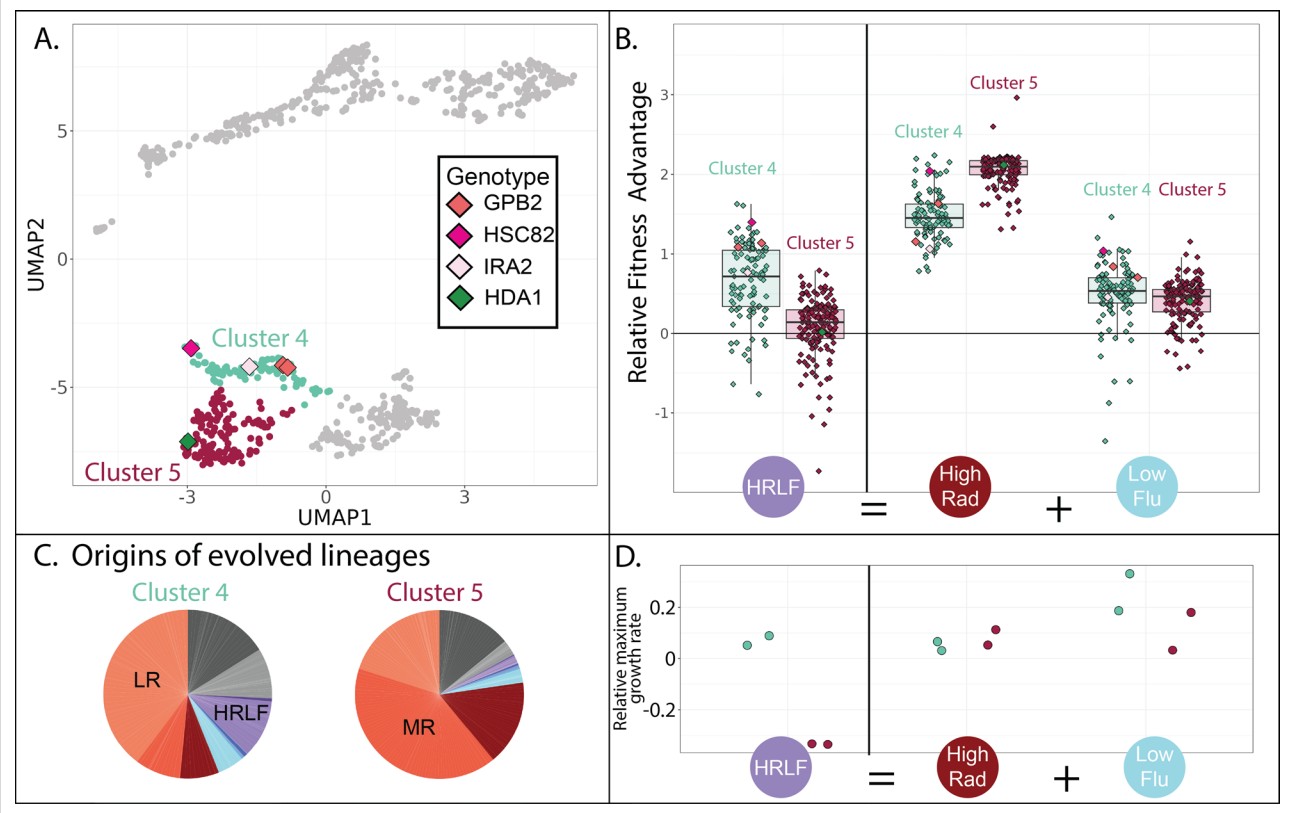

**Figure 7.** Evolved lineages in cluster 4 and 5 differ in response to combined drugs. (**A**) Adjacent clusters 4 and 5 each contain a small number of sequenced isolates depicted as diamonds; diamond colors correspond to the genes containing adaptive mutations in each sequenced isolate. (**B**) Cluster 5 (red) has an unexpected fitness disadvantage in the HRLF multidrug environment relative to cluster 4 (green), given that cluster 5 lineages do not have a fitness disadvantage in the relevant single drug environments. Boxplots summarize the distribution across all lineages within each cluster in each environment, displaying the median (center line), interquartile range (IQR) (upper and lower hinges), and highest value within 1.5 × IQR (whiskers). (**C**) Pie charts display the relative frequency with which lineages in each cluster were sampled from each of the 12 evolution conditions, colors match those in *Table 1*. (**D**) The maximum exponential growth rate for a single lineage isolated from each of clusters 4 (green) and 5 (red), relative to the ancestor. The growth rate of each lineage in each condition was measured twice by measuring changes in optical density over time. Tested lineage from cluster 4 (in green) has a mutation to GBP2 (S317T) while the lineage from cluster 5 (in red) has mutation to HDA1 (S600S).

The online version of this article includes the following figure supplement(s) for figure 7:

**Figure supplement 1.** Unexpected tradeoffs in evolved lineages in cluster 4 and 5 in response to combined drugs.

the opposite trend from those in cluster 1. They perform best in the highest concentration of FLU and have reduced fitness in lower concentrations (*Figure 5C*). These findings provide additional evidence that a distinct mechanism of FLU resistance distinguishes cluster 1 from clusters 2 and 3.

To confirm that different drug-resistant mutants dominate evolution in just slightly different concentrations of the same drug, we used each cluster's barcodes to trace mutants back to the evolution experiment from which they originated. Mutants in cluster 1 predominantly originated in evolutions containing the lowest concentration of FLU (*Figure 5A*; pie charts), while mutants in clusters 2 and 3 more often originated from evolution experiments containing higher FLU concentrations (see *Figure 6*). We also confirmed that the mutants in cluster 1 have high fitness in low FLU only by measuring growth curves for SUR1, UPC2, and PDR mutants in three concentrations of fluconazole. Lineages from cluster 1 (SUR1, UPC2) indeed grow faster and reach higher density in low FLU than those from cluster 3 (PDR; *Figure 5D*). But lineages from cluster 1 (SUR1, UPC2) grow poorly in higher FLU concentrations, while lineages from cluster 3 (PDR) do not suffer this tradeoff (*Figure 5E*). The observation that different mutants acting via different resistance mechanisms dominate evolution in only slightly different concentrations of the same drug highlights the complexity of adaptation and the potential benefits of more deeply understanding the diversity of adaptive mechanisms before designing treatment strategies (*Berman and Krysan, 2020*; *Yang et al., 2023*).

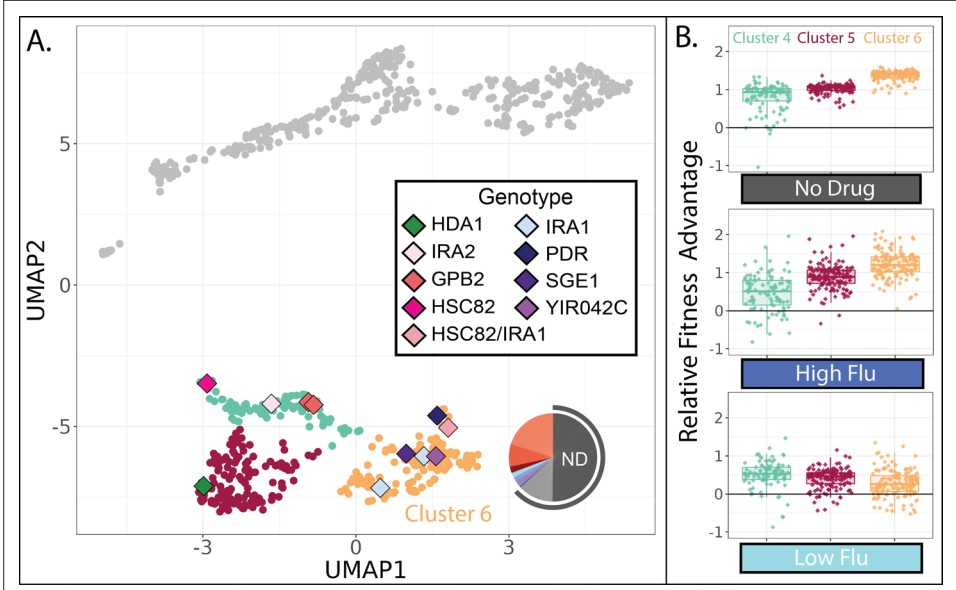

**Figure 8.** Evolved lineages in cluster 6 have higher fitness than other lineages in the absence of FLU and RAD. (**A**) Same UMAP as *Figure 4D* with clusters 4, 5, and 6 highlighted and sequenced isolates in these clusters represented as diamonds. Diamond colors correspond to the targets of adaptation in the sequenced isolates. Pie charts display the relative frequency with which lineages in cluster 6 were sampled from each of the 12 evolution conditions; colors match those in *Table 1*. Grey outline depicts conditions lacking RAD and FLU. (**B**) Of the three clusters on the bottom half of the UMAP, cluster 6 lineages perform best in conditions without any drug and in the highest concentration of FLU. Yet they perform worst in the lowest concentration of FLU. Boxplots summarize the distribution across all lineages within each cluster in each environment, displaying the median (center line), interquartile range (IQR) (upper and lower hinges), and highest value within 1.5×IQR (whiskers).

## Two groups of mutant lineages possessing similar adaptive mutations differ in sensitivity to RAD

While cluster 1 appears fairly different from its neighbors, it is not immediately obvious why the mutant lineages in clusters 2 and 3 are placed into separate groups. For one, the mutants in clusters 2 and 3 have fitness profiles with a very similar shape (*Figures 4E and 6A*). The sequenced lineages in each of these clusters also possess mutations to the same genes: PDR1 and PDR3 (*Figure 5A*). And finally, the lineages in each cluster originate from similar evolution experiments, largely those containing FLU (*Figure 6A*; pie charts). These observations made us wonder whether the difference between cluster 2 and 3 arose entirely because the mutants in cluster 3 have stronger effects than those in cluster 2 (*Figure 6A*; the solid blue line is above the solid orange line). In other words, we wondered whether the mutant lineages in clusters 2 and 3 affect fitness via the same mechanism, but to different degrees. To investigate this idea, we normalized all fitness profiles to have the same height on the vertical axis; this does not affect their shape (*Figure 6A*; dotted lines). Then we re-clustered and asked whether mutants pertaining to the original clusters 2 and 3 were now merged into a single cluster. They were not (*Figure 6B*).

Normalizing in this way did not radically alter the UMAP, which still contains largely the same six clusters of mutants (*Figure 6—figure supplement 1*). Clusters 2 and 3, containing lineages with mutations to PDR1 or PDR 3, experienced the largest changes with 37% of mutants switching from one of these two groups to the other. The new clusters 2 and 3 now differ in the shape of their fitness profiles, whereby slight differences that existed between the original fitness profiles are exaggerated (*Figure 6B*). For example, mutants in cluster 3 perform better in high and medium concentrations of RAD (*Figure 6B*; line plot). This difference in fitness is reflected in the evolution experiments, with more mutant lineages in cluster 3 originating from the evolutions performed in RAD (*Figure 6B*; pie charts). Though cluster 3 mutants tend to have stronger RAD resistance, they tend to have reduced fitness in conditions containing neither FLU nor RAD as compared to cluster 2 lineages (*Figure 6B*; line plot). In sum, the differences between lineages in clusters 2 and 3 were not resolved upon normalizing

fitness profiles to reduce magnitude differences, instead they were made more apparent (*Figure 6*). These differences do not appear to be random because they persist across independent experiments. For example, cluster 3 mutants are more fit in both medium and high RAD environments (*Figure 6B*; line plot) and were more often isolated from evolutions containing RAD (*Figure 6B*; pie charts). The observation that PDR mutations fall into two separate clusters begs a question: how can different mutations to the same gene affect fitness via different molecular mechanisms?

Asking this question forces us to consider what we mean by 'mechanism'. The mechanism by which mutations to PDR1 and PDR3 affect FLU resistance is well established: they increase transcription of an efflux pump that removes FLU from cells (*Buechel and Pinkett, 2020*; *Moye-Rowley, 2019*; *Osset-Trénor et al., 2023*). But if this is the only molecular-level effect of mutations to these genes, it is difficult to reconcile why PDR mutants fall into two distinct clusters with differently shaped fitness profiles. Others have also recently observed that mutants to PDR1 do not all behave the same way when exposed to novel drugs or changes in pH (*Chen et al., 2023*). This phenomenon is not reserved to PDR mutants, as adaptive missense mutations to another gene, IRA1, also do not share similarly shaped fitness profiles either (*Kinsler et al., 2020*). One explanation may be that, while all adaptive mutations within the same gene improve fitness via the same mechanism, not all mutants suffer the same costs. For example, perhaps the adaptive PDR mutations in cluster 2 cause misfolding of the PDR protein, resulting in lower fitness in RAD because this drug inhibits a chaperone that helps proteins to fold. In this case, it might be more correct to say that each of our six clusters affects fitness through a different, but potentially overlapping, suite of mechanisms (*Wang et al., 2023*). Previous work demonstrating that mutations commonly affect multiple traits supports this broader view of the mechanistic differences between clusters (*Boyle et al., 2017*; *Geiler-Samerotte et al., 2020*; *Kinsler et al., 2020*; *Paaby and Rockman, 2013*).

Alternatively, perhaps not all adaptive mutations to PDR improve fitness via the same mechanism. PDR1 and PDR3 regulate transcription of YOR1 and SNQ2 as well as PDR5, and maybe the different clusters we observe represent mutants that upregulate one of these downstream targets more than the other (*Osset-Trénor et al., 2023*). Or, the mutants in each cluster might harbor different aneuploidies or small, difficult to sequence chromosomal insertions or deletions that affect fitness. We leave identification of the precise mechanisms that differentiate these clusters for future work. Here, using the example of PDR mutants, we showcase how genotype may not predict fitness tradeoffs, suggesting there is more to learn about the mechanisms underlying FLU resistance.

## One group of RAD resistant mutants does not respond as expected to drug combinations

Although the three clusters of mutants on the bottom half of the UMAP are all advantageous in RAD and in conditions without any drug (*Figure 4E*; lower three plots), they differ in their fitness in conditions containing FLU. For example, the cluster of yeast lineages highlighted in green (cluster 4 in *Figures 4 and 7A*) is unique in that it has a slight advantage in the HRLF environment (*Figure 7B*). We found it especially strange that the neighboring cluster 5 does not also have a fitness advantage in this condition. Mutants in cluster 5 have a slight advantage in the LF condition, and a big advantage in the HR condition, thus we expect them to have at least some fitness advantage in the condition where these two drugs are combined (HRLF), but they do not (*Figure 7B*). The same is true for the combination of LRLF: cluster 5 mutants have an advantage in both single drug conditions which is lost when the drugs are combined (*Figure 7—figure supplement 1*). However, the mutants in cluster 4 (green) exhibit no such sensitivity to combined treatment. They have a slight advantage in all of the aforementioned single drug conditions, which is preserved in the relevant multidrug conditions (*Figure 7B*, *Figure 7—figure supplement 1*). To obtain an independent measure of the fitness of cluster 4 vs. cluster 5 lineages in these multidrug conditions, we asked from where the lineages in each cluster originate. About 10% of cluster 4 lineages originated from the HRLF evolution, while almost none of the lineages in cluster 5 came from this experiment, confirming that cluster 5 lineages are uniquely sensitive to this multidrug environment (*Figure 7C*).

The different fitness profiles of mutants in cluster 4 vs 5 (*Figure 7B*, *Figure 7—figure supplement 1*) might imply that they have different growth phenotypes. We performed a follow-up experiment that supports this observation. We asked whether there are differences in the growth phenotypes of cluster 4 vs 5 mutants by measuring a growth curve for the lineage we were able to isolate and

sequence from cluster 5, comparing it to a growth curve from a cluster 4 lineage (*Figure 7D*). Indeed, the selected mutants in cluster 4 and 5 appear to have markedly different growth curves in some conditions (*Figure 7—figure supplement 1*). The growth differences echo those we see in the fitness data. For example, the cluster 5 mutant has a lower maximum growth rate in the HRLF multidrug condition, corresponding with their lower fitness in this condition (*Figure 7B and D*). The cluster 5 mutant also reaches a lower maximum cell density in the LRLF multidrug condition and also has lower fitness in this condition (*Figure 7—figure supplement 1*). However, the growth curves and fitnesses of cluster 4 and 5 mutants are more similar in LF, LR, and HR single drug conditions. The observation of reproducible growth differences between a cluster 4 and cluster 5 mutant provides some supporting evidence that our clustering approach is effective at separating mutants with different properties.

## One group of RAD resistant mutants is exceptionally adaptive in conditions without drug

One group of mutants in the lower half of the UMAP (cluster 6 in *Figure 8A*) appears distinct from the other two in that it has the largest fitness advantage in conditions lacking any drug (*Figure 8B*). This might imply that cluster 6 lineages rose to high frequency during our evolution experiments in environments without either drug, specifically the 'no drug' and 'DMSO' control conditions. Indeed, this is what we observe: over 50% of the lineages in cluster 6 were sampled from one of these two evolution experiments (*Figure 8A*). On the contrary, the other clusters in the lower half of the UMAP consist mainly of lineages sampled from one of the RAD evolutions (*Figure 7C*). Since our fitness experiments were performed independently of the evolution experiments, this provides two independent pieces of evidence suggesting that lineages in cluster 6 are defined by their superior performance in conditions lacking any drug.

In line with the success of cluster 6 mutants in no drug conditions, the five sequenced mutants in this cluster include three that have mutations to IRA1, which was the most common target of adaptation in another evolution experiment in the condition we call 'no drug' (*Figure 8A*; *Venkataram et al., 2016*). In that experiment, and in our no drug experiment, mutations to IRA1 result in a greater fitness advantage than mutations to its paralog, IRA2, or mutations to other negative regulators of the RAS/PKA pathway such as GPB2 (*Figure 8*). Previous work showed that sometimes IRA1 mutants have very strong tradeoffs, for example, they become extremely maladaptive in environments containing salt or benomyl (*Kinsler et al., 2020*). We do not observe this to be the case for either FLU or RAD. In fact, we observe that cluster 6 mutants, including those in IRA1, maintain a fitness advantage in our highest concentration of both drugs (*Figure 4*), being more fit in high FLU than mutants in either of the other clusters in the lower half of the UMAP (*Figure 8B*). However, cluster 6 mutants are unique in that they lose their fitness advantage in the lowest concentration of FLU (*Figure 8B*). Being singularly sensitive to a low concentration of drug seems unusual, so much so that when this was observed previously for IRA1 mutants the authors added a note about the possibility of a technical error (*Kinsler et al., 2020*). Our results suggest that there is indeed something uniquely treacherous about the low fluconazole environment, at least for some genotypes.

## Discussion

Here, we present a barcoded collection of fluconazole (FLU) resistant yeast strains that is unique in its size, its diversity, and its tractability. One way we were able to isolate diverse types of FLU-resistance was by evolving yeast to resist diverse drug concentrations and combinations. But the more important tool used to increase both the number and type of mutants in our collection was DNA barcodes. These allowed us to sample beyond the drug resistant mutants that rise to appreciable frequency and to collect mutants that would eventually have been outcompeted by others. Our primary goal in collecting these mutants was to get a rough sense of how many different mechanisms of FLU resistance may exist. This question is relevant to evolutionary medicine (because more mechanisms of resistance make it harder to design strategies to avoid resistance), evolutionary theory (because more mechanisms of adaptation make it harder to predict how evolution will proceed), and genotype-phenotype mapping (because more mechanisms makes for a more complex map).

We distinguish mutants that likely act via different mechanisms by identifying those with different fitness tradeoffs across 12 environments, leveraging the mutants' barcodes to track their relative

fitness following previous work (*Kinsler et al., 2020*). The 774 FLU-resistant mutants studied here clustered into a handful of groups (6) with characteristic tradeoffs. We confirmed that each group captures mutants with distinct properties using multiple approaches, including whole genome sequencing, growth curve experiments, tracing the evolutionary origins of the mutants in each cluster, and by using two additional independent clustering methods: hierarchical clustering and PCA (*Figures 5–8*). Some groupings are unintuitive in that they segregate mutations within the same gene (*Figure 6*) or are distinguished by unexpectedly low fitness in multidrug conditions (*Figure 7*). These findings are important because they challenge strategies in evolutionary medicine that rely on consistent tradeoffs or intuitive trends when designing sequential drug treatments. On the other hand, the observation that some mutants have very similar tradeoffs such that they cluster together is promising in that it suggests predicting the impact of some mutations by understanding the impacts of others is somewhat feasible.

Problematically, the clusters we present are incomplete and bound to change as additional data presents itself. For one, we have shown that additional FLU-resistant mutants emerge from evolution experiments in conditions lacking FLU (*Figure 3C and D*). This begs questions about what other FLU-resistant mutants might emerge in environments we have not studied here. Additionally, previous work has shown that some mutants that group together in our study (e.g. GPB2 and IRA2) have different fitness profiles in conditions that we did not include here (*Kinsler et al., 2020*). Also of note is that our evolution experiments were conducted for only a few generations and all started from the same genetic background. Additional types of FLU-resistant mutants with unique fitness profiles may emerge from other genetic backgrounds or arise after more mutations are allowed to accumulate (*Allen et al., 2021*; *Bosch et al., 2021*; *Brandis et al., 2012*). Finally, by requiring that all included mutants have sufficient sequencing coverage in all 12 environments, our study is underpowered to detect adaptive lineages that have low fitness in any of the 12 environments. This is bound to exclude large numbers of adaptive mutants. For example, previous work has shown some FLU resistant mutants have strong tradeoffs in RAD (*Cowen and Lindquist, 2005*). Perhaps we are unable to detect these mutants because their barcodes are at too low a frequency in RAD environments, thus they are excluded from our collection of 774. All of the aforementioned observations combined suggest that there are more unique types of FLU-resistant mutations than those represented by these 6 clusters, and that the molecular mechanisms that can contribute to fitness in FLU are more diverse than we know. This could complicate (or even make impossible) endeavors to design antimicrobial treatment strategies that thwart resistance.

On the up side for evolutionary medicine, not every infection harbors all possible types of mutants. This might explain why strategies that exploit one or two common tradeoffs have some, albeit mixed, success in delaying or preventing the emergence of resistance (*Amin et al., 2015*; *Imamovic et al., 2018*; *Kaiser, 2017*; *Krishna et al., 2022*; *Nyhoegen and Uecker, 2023*; *Waller et al., 2023*; *Wang et al., 2019*). Our results encourage more complex strategies to thwart drug resistance (*Iram et al., 2021*), such as those that focus on advance screening to determine the resistance mechanisms that are present (*Andersson et al., 2019*), or on cycling a larger number of drugs to exploit a larger number of tradeoffs (*Thomas et al., 2022*; *Yoshida et al., 2017*). Problematically, these strategies often rely on knowledge about the diversity of mutants and tradeoffs that exist (or that can emerge) within an infectious population. This type of information about population heterogeneity, heteroresistance, and substructure is expensive and arduous to obtain (; *Bottery et al., 2021*). Fortunately, new methods, in addition to the one presented in this study, are emerging (*Aissa et al., 2021*; *Brettner et al., 2024*; *Forsyth et al., 2021*; *Hsieh et al., 2022*; *Kuchina et al., 2021*; *Nagasawa et al., 2021*). The richer data provided by these methods dovetails with emerging population genetic models that predict the likelihood of resistance to a given drug regimen (*Cannataro et al., 2018*; *Day et al., 2015*; *Feder et al., 2021*; *King et al., 2022*; *Read and Huijben, 2009*; *Somarelli et al., 2020*; *Wilson et al., 2016*). In sum, our observation of numerous different types of drug-resistant mutations suggests that designing resistance-detering therapies is challenging, but perhaps not impossible.

Outside of predicting the evolution of resistance, our findings provide a tool to investigate the phenotypic impacts of mutation. This task has proven daunting in light of work demonstrating that mutations often have many phenotypic impacts (*Boyle et al., 2017*; *Paaby and Rockman, 2013*) and that these impacts change with contexts including the environment (*Eguchi et al., 2019*; *Geiler-Samerotte et al., 2020*; *Geiler-Samerotte et al., 2016*; *Lee et al., 2019*; *Paaby et al., 2015*). The

approach presented in this study provides a way forward by identifying mutations that cluster together such that the effects of some mutants can be predicted from others. This clustering strategy can assist high-throughput efforts to identify the phenotypic impacts of a large panel of mutations (*Flynn et al., 2024*; *Fowler and Fields, 2014*; *Mehlhoff et al., 2020*; *Starr et al., 2017*). Further, our approach identifies environments that differentiate one cluster of mutants from another. This suggests where to look to understand the phenotypes that differentiate each cluster of mutants. For example, we were able to show that the growth phenotypes of mutants from clusters 4 and 5 are different because we knew to look for these differences in multidrug environments (*Figure 7D*). And our results suggest radicicol environments may be most helpful in teasing out any phenotypic differences that set apart some PDR mutations from others (*Figure 6*). Thus, our approach guides efforts to understand the phenotypic effects of mutation, while also guiding efforts to predict the effects of some mutations from others as well as efforts to predict the outcomes of adaptive evolution.

## Methods

### Base yeast strains

All of the budding yeast (*Saccharomyces cerevisiae*) lineages studied here originated from the same starting strain referred to as the 'landing pad strain' (SHA185) in previous work (*Levy et al., 2015*). We transformed a barcode library (generously provided by Sasha Levy) into this strain as described below, creating a strain with the following genetic background:

MATα, ura3Δ0, ybr209w::Gal-Cre-KanMX-1/2URA3-loxP-Barcode-1/2URA3-HygMX-lox66/71.

### Base media

All experiments were conducted in 'M3' media defined in the same study as the landing pad strain (*Levy et al., 2015*), which is a glucose-limited media lacking uracil. In our study, we supplemented this media with fluconazole, radicicol, or DMSO when appropriate.

### Selecting drug concentrations

Our goal was to choose concentrations of each drug that would not kill so many yeast cells as to dramatically decrease barcode diversity. We wanted to maintain a high number of unique barcodes so we could track a high number of yeast lineages as they independently evolved drug resistance. We measured the effect of each drug and drug combination on the growth rate of a single barcoded yeast strain using a plate reader to track changes in optical density (OD) over time. Ultimately, we chose a 'low' concentration of each drug that appeared to have no effect on growth rate, and a 'high' concentration that appeared to reduce growth rate by about 15% (*Figure 2—figure supplement 2*). Although the lowest concentration of radicicol that we tested on a plate reader was 10 µM, we chose 5 µM as our low RAD concentration because previous work suggested this concentration had widespread effects on yeast physiology without affecting growth (*Geiler-Samerotte et al., 2016*; *Jarosz and Lindquist, 2010*). To perform our plate reader experiment, a single colony was grown to saturation. From this culture, 5 µl was added to every well of a 96-well plate, where every well contained 195 µl of M3 media. Some wells also contained either fluconazole, radicicol, DMSO, or combinations of these drugs. The concentrations that were tested are listed on the horizontal axis of *Figure 2—figure supplement 2*; each drug condition was replicated six times. The 96-well plate was incubated at 30 °C for 48 hr on a plate reader and OD measurements were taken every 30 min. Raw OD values were exported and maximum exponential growth rates for all tested conditions were calculated from the log-linear changes in OD over time.

### Inserting 300,000 unique DNA barcodes into otherwise genetically identical yeast cells

In order to track many yeast lineages as they independently develop drug resistance, we needed to insert unique DNA barcodes into many yeast cells. Plasmids harboring barcodes (pBar3) were the same as those used in a previous barcoded evolution experiment (*Levy et al., 2015*) and were generously provided to us by Sasha Levy. These barcodes are 25 base pairs in length. They are targeted to an artificial intron within the Ura3 gene, such that they must be retained in media lacking uracil but are not expressed and thus do not themselves affect fitness (*Levy et al., 2015*). We transformed this

barcode library (pBar3) into the landing pad strain (SHA185) as was done previously, activating a Cre-lox recombination system by growing the cells in YP-galactose, which resulted in genomic integration of the barcode. However, our efforts to perform extremely high efficiency transformations from which we could isolate hundreds of thousands of uniquely barcoded yeast were unsuccessful, despite manipulating the levels and timing of the inducer (galactose). Ultimately we performed 24 separate transformations and pooled many of these to obtain a large pool of barcoded yeast where every yeast cell was genetically identical except for its DNA barcode.

## Examining the frequency of each barcode in the starting pool of cells

We sequenced the barcode region of these 24 transformed yeast populations on the Hiseq X platform using a dual index system (*Kinsler et al., 2023*) to discern barcode coverage, that is how many total unique barcodes were successfully inserted into yeast cells and how evenly these barcodes were sampled. We needed many uniquely barcoded yeast in order to observe many different adaptive lineages within each evolution experiment. But barcodes with very high frequencies, referred to herein as monster lineages, were present in 10 of the 24 transformations and present a problem. Monster lineages allow too many cells to carry the same barcode, giving that barcode more chances to develop an adaptive mutation. This could allow different cells harboring that same barcode to pick up different adaptive mutations, destroying our ability to draw conclusions about adaptive mutations by using barcodes. Therefore, our final library of barcoded lineages was created by pooling 14 individual transformations together, choosing those 14 that lacked monster lineages, which we defined as lineages representing greater than 1% of all transformants. Our sequencing results suggest that this library contains about 300,000 unique barcodes.

## Initiating 12 barcoded evolution experiments

All evolution experiments started from the same pool of roughly 300,000 uniquely barcoded yeast lineages. To start the evolution experiments, a pea sized amount of the frozen yeast barcode library was grown up in 4 ml YPD for 4 hr at 30 °C in a shaking incubator at 220 rpm. Then, 300 μl of the grown barcode library was added to each of 12 pre-prepared 500 ml flasks representing the 12 evolution experiments listed in *Table 1*. To prepare these flasks, first, 1.2 l of M3 media was warmed at 30 °C. Then, 100 ml was added to each of 12 flat bottom flasks. Next, 500 μl of the appropriate drug or drug combination was added to each flask. Drugs were pre-diluted, aliquoted and frozen such that 500 μl of the appropriate tube could be added to each flask to achieve the desired concentration as listed in *Table 1*. All drugs were resuspended in DMSO such that the final concentration of DMSO in all experiments (except the 'no drug' control) was 0.5%.

## Performing barcoded evolution experiments

Evolution experiments were performed following previous work (*Levy et al., 2015*). After initiation (see above), the yeast in every flask were allowed to grow at 30 °C with shaking at 200 RPM for 48 hr. Then, the flasks were removed from the incubator and 400–1000 μl of each culture was transferred to a new pre-prepared flask with identical conditions to the first. The reason we added more volume (1000 μl) to some flasks than previous work was that the cell counts at the end of the 48 hr were lower for some of our higher drug conditions. We adjusted the transfer volume to maintain a transfer population of $4\times10^7$ cells, which was the same as in previous work (*Levy et al., 2015*). We completed a total of 24 growth/transfer cycles, corresponding to 192 generations of growth assuming 8 generations per 48 hr cycle (*Levy et al., 2015*). Following each transfer, the remaining culture from each flask were split into two 50 ml conical vials, centrifuged for 3 min at 4000 rpm, and the supernatant was discarded. The final pellet was resuspended in 30% glycerol up to a total volume of 6 ml before being split into three 2 ml cryovials and stored at –80 °C. These frozen samples were later utilized for barcode sequencing and isolating adaptive mutants.

## Isolating a large pool of adaptive mutants

To generate a large pool of diverse adaptive mutants, our goal was to collect a sample from each evolution experiment at a time point when there were many different adaptive lineages competing. If we sampled too late, the adaptive lineage with the greatest fitness advantage would have already risen to high frequency, thus reducing diversity. But if we sampled too early, adaptive lineages would

not yet have risen in frequency above other lineages. Therefore, we chose to sample cells from a time in each evolution experiment when many barcoded lineages appeared to be rising in frequency (*Figure 2—figure supplement 1*). We sampled either 1 or 2 thousand cells per each evolution experiment by spreading frozen stock from the chosen time point onto agarose plates, scraping 1 or 2 thousand colonies into a 15 mL conical tube containing a final concentration of 30% glycerol, and freezing the pool pertaining to each of the 12 evolutions. We sampled 2000 cells from most evolution experiments, but sampled only 1000 from those containing a high concentration of FLU as those evolutions appeared to have reduced barcode diversity (*Figure 2—figure supplement 1*), presumably because high FLU represents a strong selective pressure. We sequenced the barcodes from each of these 12 pools so that we could track which adaptive mutants originated from which evolution experiment (see Methods section below entitled, 'Inferring where adaptive lineages originally evolved').

## Initiating barcoded fitness competition experiments

To assess the fitnesses of the 1 or 2 thousand barcoded lineages that we sampled from each evolution experiment, we pooled all sampled lineages together into a larger pool of roughly 21,000 barcoded lineages. We used this larger pool to initiate 24 fitness competition experiments, 2 replicates for each of the 12 conditions listed in *Table 1*. In this type of competition, we measure fitness by tracking changes in each barcode's frequency over time. Barcodes that rise in frequency represent strains that have higher fitness than others.

Our goal was to calculate the fitness effect of adaptive mutations. Therefore, we needed to calculate the fitness of every evolved lineage relative to the unmutated ancestor of the evolution experiments. To do so, we followed previous work by spiking in a large quantity of this unmutated ancestor strain into each fitness competition, with this ancestor making up at least 90% of the final culture (*Kinsler et al., 2020*; *Venkataram et al., 2016*). In environments containing a high concentration of FLU which resulted in the ancestral strain having a more severe growth defect, we spiked in the ancestor such that it represented 95% of the final pool.

To avoid wasting 90% or more of our sequencing reads on the ancestor strain's barcode, we created a barcodeless ancestor strain. This strain was created by transforming SHA185 with a linear piece of DNA such that the genetic background was identical to the strains of the barcoded library, but the homology to the primers used to amplify the barcode was missing. Thus the DNA from these cells does not get amplified or sequenced during subsequent steps.

In addition to this barcodeless ancestor, we also spiked in some barcoded ancestral strains at lower frequency (1%) to use as 'reference' or 'control' strains, following previous work (*Kinsler et al., 2023*; *Kinsler et al., 2020*). These strains have been previously shown to possess no fitness differences from the ancestor. We used these strains as a baseline when calculating relative fitness by setting the fitness of these strains to zero during our fitness inference procedure (see Methods section below entitled, 'Inferring fitness').

All 24 fitness competitions were performed simultaneously in one big batch (*Kinsler et al., 2023*) and initiated from the same pool of roughly 21,000 barcoded evolved yeast lineages, barcodeless ancestor, and control strains. To initiate the competitions, $7\times10^7$ cells from this pool were added to 24 pre-prepared 500 mL flasks corresponding to the conditions listed in *Table 1*. These flasks were prepared exactly the same way as was done for the evolution experiments (see above in 'Performing barcoded evolution experiments'). Each flask was allowed to grow for 48 hours at 30 °C with shaking at 200 RPM.

## Performing barcoded fitness competition experiments

Fitness competitions were performed following previous work (*Kinsler et al., 2020*). After the initial flasks were allowed to grow for 48 hours, they were removed from the incubator and 400 µl from each culture representing $4\times10^7$ cells were transferred to a new flask with identical media. For each of 24 competitions, we completed a total of 4 growth/transfer cycles, corresponding to 40 generations of growth assuming 8 generations per 48 hr cycle (*Levy et al., 2015*). Following each transfer, the remaining culture from each flask was split into two 50 ml conical vials, centrifuged for 3 min at 4000 rpm, and the supernatant was discarded. The final pellet was resuspended in 30% glycerol up to a total volume of 6 ml before being split into three 2 ml cryovials and stored at –80 °C. These frozen samples were later utilized for DNA extraction and subsequent barcode sequencing.

Despite the fitness competition experiments being conducted for nearly the same number of generations (40) as were the evolution experiments before isolating adaptive lineages, we do not anticipate that secondary mutations will bias fitness measurements. Previous work has demonstrated that in this evolution platform, most mutations occur during the transformation that introduces the DNA barcodes (*Levy et al., 2015*). In other words, these mutations are already present and do not accumulate during the 40 generations of evolution. Therefore, the observation that we collect a genetically diverse pool of adaptive mutants after 40 generations of evolution is not evidence that 40 generations is enough time for secondary mutations to bias abundance values. For a detailed treatment of how secondary mutations have a minimal influence on fitness, see *Venkataram et al., 2016*.

## Extracting genomic DNA

DNA was extracted from 500 μl of concentrated frozen stocks pertaining to the evolution experiments and fitness competitions. Frozen cells were thawed and pelleted. Cells were treated with 250 μl of 0.1 M Na2EDTA, 1 M sorbitol and 5 U/μl zymolyase for a minimum of 15 min at 37 °C to remove the cell wall. Lysis was completed by adding 250 μl of 1% SDS, 0.2 N NaOH and inverting to mix. Proteins and cell debris were removed with 5 M KOAc by spinning for 5 min at 15,000 rpm. Supernatant was moved to a new tube and DNA was precipitated with 600 μl isopropanol by spinning for 5 min at 15,000 rpm. The resulting pellet was washed 1 ml of 70% ethanol before being resuspended in 50 μl water plus 10 μg/ml RNAse. Extracted DNA was quantified using the NanoDrop spectrophotometer and all samples were diluted to a concentration of 50 ng/μl for barcode amplification and sequencing library preparation.

## Preparing barcodes for high-throughput multiplexed sequencing using PCR

Extracted DNA was prepared for sequencing using a two-step PCR that preserves information about the relative frequency of each barcode in each sample (*Kinsler et al., 2023*; *Kinsler et al., 2020*; *Venkataram et al., 2016*). Briefly, in the first step PCR, the barcode region is amplified from the genomic DNA, labeled with a sample-specific combination of primers, and tagged with a UMI. This step utilizes a short 3 cycle PCR with New England Biolabs OneTaq polymerase. Purification of the first step product to remove excess reagents was performed using Thermo Scientific GeneJET PCR Purification Kit. The second step PCR attached Illumina indices that were used to distinguish samples from different experiments and timepoints. We utilized a dual indexing scheme to prevent index misassignment that is common when sequencing amplicon libraries using patterned flow cell technology (*Kinsler et al., 2023*). Amplification of this second step of PCR was done with a longer 23 cycle PCR using Q5 polymerase. Final libraries were bead purified using 0.8 X Quantabio sparQ Pure Mag beads. Quantification of the final PCR products was done using the Invitrogen Qubit Fluorometer before all samples were pooled at equimolar ratios for sequencing.

## Sequencing and clustering barcodes

Next Generation Sequencing was performed at either Psomagen (Rockville, MD) or at the Translation Genomics Research Institute (Phoenix, AZ) on patterned flow cells (either an Illumina HiSeqX or NovaSeq) using 2x150 base pair paired end reads. Samples were dual indexed to allow multiplexing while minimizing contamination from index misassignments (*Kinsler et al., 2023*). The 20 base pairs of variable sequence referred to as a DNA barcode were identified and clustered to determine the number of unique barcodes and the frequency of each barcode in each sample. For the evolution experiments, this was done following our previous work (*Kinsler et al., 2020*; *Venkataram et al., 2016*). For the fitness competition experiments, this was done using updated software (*Zhao et al., 2018*) with the following command:

## Inferring fitness

In fitness competition experiments, fitness is often inferred from the log-linear change in a strain's frequency over time (*Bakerlee et al., 2021*; *Geiler-Samerotte et al., 2011*; *Kinsler et al., 2023*). Recently, more advanced methods to infer fitness have emerged that take into account nonlinearities in frequency changes over time, for example, nonlinearities that reflect changes in the mean fitness of the population (*Kinsler et al., 2020*; *Li et al., 2018a*; *Li et al., 2023*; *Venkataram et al., 2016*). We

had trouble implementing these newer methods on our fitness data, perhaps because many of our evolved lineages, and our control strains, have low fitness in some drugs. This caused their barcodes to rapidly decline in frequency such that they received low counts only at later time points. Their counts could become so low that these lineages would seemingly disappear due to sampling error, and then reappear at a subsequent time point. This dramatic (but false) late increase in frequency was sometimes interpreted as evidence of very high fitness, especially when we inferred fitness using approaches that account for nonlinearities.

To contend with this issue, we applied strict coverage thresholds to every fitness measurement: we required at least 500 counts across all timepoints in order to infer fitness for a given lineage in a given environment. This is stricter than previous work that does not require a minimum number of reads per each lineage and instead requires a minimum number of reads per time point (*Kinsler et al., 2020*). We found that 774 lineages passed our threshold in at least one replicate experiment per all 12 environments. Of these, 729 passed for both replicates and the final fitness value we report represents the average of both replicates.

Even with our strict coverage threshold, fitness inference methods that account for nonlinearities still interpreted minor stochastic fluctuations in fitness at later time points as evidence of a fitness advantage, even if fitness dramatically declined in earlier time points. Therefore, we calculated fitness via the traditional method, as the slope of the log-linear change in barcode frequency relative to the average slope of the control strains. We found that this method is less sensitive to that type of error. Using this method, we found that our fitness inferences were reproducible between replicates (*Figure 2—figure supplement 4A*), and between experiments performed in similar conditions (e.g. medium vs. high concentrations of the same drug; *Figure 2—figure supplement 4B*). When we increased our coverage threshold to require an order of magnitude more reads per lineage per measurement (from 500 to 5000), we lost 157 lineages (from 774 to 617), saw reproducibility increase across replicates (from an average Pearson correlation of 0.756–0.813) and the main conclusions of our study were unchanged in that the same 6 clusters were present on a UMAP (*Figure 4—figure supplement 2*).

## Identifying adaptive mutations using whole-genome sequencing

One downside of barcoded evolution experiments is that all lineages exist together in a pooled culture. Fishing out adaptive lineages in order to perform whole genome sequencing is a major challenge (*Venkataram et al., 2016*). Here, we randomly selected cells from these mixed pools for whole genome sequencing, sometimes selecting from later time points in the evolution experiments and sometimes selecting from the samples of 1 or 2 thousand cells that were isolated to initiate fitness competitions.

To perform whole genome sequencing, cells from mixed pools were spread onto M3 agarose plates, single colonies were selected and grown in YPD to saturation. DNA was extracted using the PureLink Genomic DNA Mini Kit (K182002). Sequencing libraries were made using Illumina DNA Prep kit by diluting reactions by ⅕. Briefly, samples were prepared such that the starting concentration in 6 µl was between 20 and 100 ng of DNA. 2 µl of BLT and TB1were added to the starting material and incubated on a thermocycler at 55 °C (lid 100 °C) for 15 min. Two µl of TSB was added to each reaction and incubated at 37 C (lid 100 °C) for 15 min. Beads were washed two times with 20 µl of TWB. Following the final wash, 4 µl of EPM, 4 µl of water and 2 µl of UD indexes were added to each sample. Depending on starting concentration, PCR was performed based on Illumina guidelines as follows: lid 100 °C, 68 °C for 3 min, 98 C for 3 min, [98 °C for 45 s, 62 °C for 30 s, 68 °C for 2 min] for 6–10 cycles, 68 °C for 1 min, 10 °C hold. PCR products were cleaned with a double side sized selection as follows: 4 µl of each sample was pooled together (32 µl total for 8 samples) and added to 28 µl of water plus 32 µl of SPB. After a 5 min incubation, 25 µl of supernatant was moved to a new tube containing 3 µl of SPB. Beads were washed with fresh 80% ethanol and libraries were eluted in 12 µl RSB. Samples were multiplexed using Illumina's unique dual (UD) index plates (A-D) and sequencing was performed with 2x150 paired end sequencing on HiSeq X at Psomagen (Rockville, MD).

In total 122 colonies were randomly picked and sequenced. As one might expect, barcodes that rose to high frequency were more likely to be picked multiple times. In an attempt to find lineages with unique attributes, some cultures were grown at 37 °C or plated to high concentrations of drug prior to picking isolated colonies for sequencing. Of the 122 genomes we sequenced, only 53 had unique

barcodes that pertained to the 774 lineages for which we obtained high enough barcode coverage to infer fitness. Only two of these 53 had no sequenced mutations suggesting their fitness increase over ancestor is due to a mutation we are unable to identify by sequencing, perhaps a change in ploidy. The other 51 all had at least one single nucleotide mutation in a gene reported in *Supplementary file 1*. Whole genome sequences were deposited in GenBank under SRA reference PRJNA1023288.

Variant calling was done using GATK as described here: https://github.com/gencorefacility/variant-calling-pipeline-gatk4 (*Khalfan, 2020*). Identified variants were annotated using SnpEff (*Cingolani et al., 2012*). Variant call files from 132 (53 unique/in CS) sequenced lineages were analyzed in R and compared to reference strain GCF_000146045.2 (Genome assembly 64: sacCer3). SNPs present in the ancestor (as well as all evolved lineages) were ignored as these could not have caused the fitness differences we observed. We also ignored SNPS that were present in a substantial number of evolved lineages, as these likely represent background mutations that were present in a substantial portion of the cells representing the landing pad strain (SHA185). These are reported in *Supplementary file 1* and include: SRD1-Glu97Lys, RSC30-Gly571Asp, OPT1-Val143Ile, and LYS20-Thr29Met.

## Measuring growth curves of evolved lineages with unexpected fitness in multidrug conditions

Though fitness differences are not necessarily due to differences in maximum growth rate (*Li et al., 2018b*), we measured growth curves for a few lineages. In one case, we did so to investigate a case where an evolved lineage had unexpectedly low fitness in multidrug conditions (*Figure 7B*). Indeed, we found that this mutant grew more slowly in those conditions (*Figure 7D*; *Figure 7—figure supplement 1*). To perform this test, a lineage with a mutation to GBP2, a lineage with a mutation to HDA1, and sometimes the ancestor strain were streaked to YPD plates. We used the barcodeless ancestor strain, which is identical to the evolved lineages in every way except for lacking a barcode, and is described above in the methods section entitled, 'Initiating barcoded fitness competition experiments'. A single colony of each strain was isolated from YPD plates and was used to inoculate an overnight YPD culture. After ~24 hr, a coulter counter (BD) was used to determine the number of cells/ml present in each culture. Next, all cultures were diluted such that the starting number of cells was 250,000 in 6 ml of M3 plus drug (either HR, LF, LR, LRLF, or HRLF, see *Table 1*). To measure growth curves, these samples were allowed to grow at 30° C. OD was measured every 10 min as the cultures were grown to saturation using the compact rocking incubator TVS062CA (Advantec Mfs). Raw growth curves for these conditions are shown in *Figure 7—figure supplement 1B and C*. Maximum growth rate was calculated using a sliding window approach to determine the region of each growth curve with the steepest log-linear slope. We used similar methods to measure growth curves for 3 mutants from cluster 1 and 3 from cluster 3 in *Figure 5*.

## Determining ploidy

While our barcoded yeast strain is haploid, previous studies observed that some cells diploidize during the course of evolution in M3 media and by doing so gain a fitness advantage (*Levy et al., 2015*; *Venkataram et al., 2016*). To ensure that observed fitness effects in our experiments were not largely due to the effects of diploids, we estimated the percent of diploid cells in each of our populations. We chose to make our estimates from frozen samples taken at the same time points from which we sampled 1 or 2 thousand cells to initiate fitness competitions (*Figure 2—figure supplement 1*). As such, our estimates also report on the percent of diploids that were present at the start of the fitness competitions experiments (*Supplementary file 2*).

To study ploidy, we used the nucleic acid stain SYTOX Green, which is capable of selectively staining the nucleus of fixed cells and has been shown to be optimal for use in budding yeast (*Haase, 2004*). For each of the 12 evolution experiments conditions, a small amount of freezer stock from the chosen timepoints (*Figure 2—figure supplement 1*) was plated to YPD and grown for ~48 hr. Individual colonies were picked and transferred to 96-well plates, one full plate per each condition, before being fixed with 95% ethanol for 1 hr. Plates were centrifuged at 4500 rpm and supernatant was discarded. A total of 50 µl RNase A was added to the samples at a concentration of 2 mg/ml, and the plates were then incubated for 2 hr at 37 °C. Cells were pelleted by centrifuge and the supernatant was removed, which was followed by treatment with 20 µl of the protease pepsin at a concentration of 5 mg/µl. Pepsin-treated samples incubated at 37 °C for 30 mins before centrifugation and

removal of supernatant. Finally, cells were resuspended in 50 µl TrisCL (50 mM, pH 8) and stained with 100 µl of 1 µM SYTOX Green. Known diploid and haploid strains were used as controls alongside our samples to determine the expected fluorescence of stained diploid vs. haploid cells. Analysis was performed using a ThermoFisher Attune NxT, housed in the Flow Cytometry Core Facility at Arizona State University.

## Dimensional reduction

Our fitness inference procedure resulted in a data set consisting of nearly 10,000 fitness measurements (774 lineages x 12 conditions = 9,288 fitness measurements). Dimensional reduction was performed on these data using UMAP (*McInnes et al., 2018*). Clusters of similar mutants were identified and colored using a gaussian mixed model *Fraley and Raftery, 2003*; Bayesian Information Criteria (*Figure 4—figure supplement 1*) as well as follow up genotyping and phenotyping studies (see *Figures 5–8*) were used to select the number of clusters. These analyses were performed in R; code can be found at https://osf.io/pxyv9/.

In order to prevent conditions with the most variation in fitness (e.g. high FLU) from dominating, we normalized fitness measurements from each of the 12 environments to have the same overall mean and variance (we transformed the data from every environment to have a mean of 0 and a standard deviation of 1) before performing dimensional reduction. This normalization procedure did not have a dramatic effect on the UMAP (*Figure 4—figure supplement 2A*). We also explored normalizing all data to account for magnitude differences by setting the average fitness of each lineage across all 12 environments to 0. Doing so did not significantly change the groupings present in the UMAP from those displayed in *Figure 4*, *Figure 6—figure supplement 1* other than in the ways we describe in *Figure 6*. Reducing our data set to 617 adaptive lineages with very high sequencing coverage *Figure 4—figure supplement 2B* also did not significantly affect the way that mutants cluster into groups, nor did using a different dimensional reduction algorithm altogether (*Figure 4—figure supplement 3* and see next paragraph). In short, the clustering of mutants was robust to the different decisions we made when choosing how to analyze these data.

In order to assess whether clusters identified from the UMAP are robust to alternative clustering methods, we also used hierarchical clustering to identify clusters of mutants with similar fitness profiles. First, we computed the pairwise distance of all lineages across the fitness profiles. Then, we used Ward's method from scikit-learn to iteratively cluster lineages such that the within-cluster variation is minimized (*Pedregosa et al., 2012*; *Ward, 1963*). To test the consistency of lineage clustering, we chose a pairwise cluster distance cutoff of 11, which results in the same number of clusters (7) as identified with the UMAP clustering approach used in the main text. We then compared the identity of the lineages within each of these clusters with the UMAP clusters. We found that, for most clusters, over 80% of lineages from the UMAP cluster corresponded with a unique hierarchical cluster and labeled these hierarchical clusters according to this correspondence (*Figure 4—figure supplement 3*). For UMAP cluster 1, lineages were more evenly split between two clusters. 64% of these lineages clustered together in what is labeled as hierarchical cluster 1 and 30% in hierarchical cluster 1/7 (*Figure 4—figure supplement 3*), which contains all of the control lineages that comprise UMAP cluster 7. Despite these lineages clustering more closely with control lineages than the remainder of the cluster 1, they do tend to cluster distinctly with the control lineages, suggesting they have behavior that is distinguishable from the control lineages. If we consider these cluster 1 mutants that end up in cluster 3/7 as 'mis-clustered', we find that 85% of lineages from each UMAP cluster are clustered together in the corresponding hierarchical cluster. If we consider these as 'consistently clustered', this metric increases to 90% of lineages correctly clustered. Similarly, clustering lineages using principal component analysis also largely preserved the clusters reported in *Figure 4*, *Figure 4—figure supplement 4*. Altogether, this analysis shows that the results we show are robust to alternative methods of clustering.

## Inferring where adaptive lineages originally evolved

All 774 adaptive lineages were isolated from one of the 12 evolution experiments at the timepoint indicated in *Figure 2—figure supplement 1* (see Methods section entitled, 'Isolating a large pool of adaptive mutants'). The sample we isolated from each evolution experiment was sequenced prior to

pooling. This allows us to computationally determine which barcoded lineages originated from which evolution experiment to generate the pie charts in *Figures 3, 5–8*.

If adaptive mutation arose independently during the course of each evolution experiment, it would be unlikely for any adaptive lineage we study to be present in more than one of the evolution conditions. This would make it very easy to assign each barcode to the evolution experiment from which it originated. However, this was not the case for many barcoded lineages.

Previous work explained that the transformation procedure used to insert a barcode into the landing pad of SHA185 is itself mutagenic, such that many mutations arise prior to the start of the evolution experiments (*Levy et al., 2015*). Since all our evolution experiments were started from the same pool of barcoded lineages, we thus expect that many adaptive lineages will be present in more than one condition. However, it is not expected that these adaptive lineages will be present at the same frequency in every condition; instead these frequencies change with the fitness of the mutation each lineage possesses. Therefore, when an adaptive lineage appeared in multiple conditions, we weighted its origin to reflect its frequency in each condition. In other words, adaptive lineages that were only present in the sample taken from a single evolution condition were identified and assigned a single origin condition in the pie charts in *Figures 3 and 5*; *Figure 6*; *Figure 7* and *Figure 8*. But for adaptive lineages found in the samples taken from more than one evolution condition, the proportions assigned to each origin condition in the pie charts was scaled to equal the relative frequencies of that lineage in all evolution conditions where it was observed. Associated data and code can be found here: https://osf.io/pxyv9/.

## Acknowledgements

We are grateful to Sasha Levy for providing plasmids and yeast strains, to Martin Mullis, Xianan Liu, Sasha Levy and Gavin Sherlock for advice about creating a high diversity library of barcoded yeast, Jamie Blundell, Sandeep Venkataram and Fangfei Li for discussions about how to infer adaptive lineages, and to the members of the Geiler-Samerotte lab as well as Michael Lynch for discussions about the manuscript. The authors acknowledge resources and support from the KE core facilities at Arizona State University. This work was supported by a National Institutes of Health grant R35GM133674 (to KGS), an Alfred P Sloan Research Fellowship in Computational and Molecular Evolutionary Biology grant FG-2021–15705 (to KGS), and a National Science Foundation Biological Integration Institution grant 2119963 (to KGS).

## Additional information

### Funding

| Funder | Grant reference number | Author |
| --- | --- | --- |
| National Institutes of Health | R35GM133674 | Kerry Geiler-Samerotte |
| Alfred P. Sloan Foundation | FG-2021-15705 | Kerry Geiler-Samerotte |
| National Science Foundation | BII 2119963 | Kerry Geiler-Samerotte |

The funders had no role in study design, data collection and interpretation, or the decision to submit the work for publication.

### Author contributions

Kara Schmidlin, Kerry Geiler-Samerotte, Conceptualization, Formal analysis, Supervision, Investigation, Visualization, Writing - original draft, Writing - review and editing; Sam Apodaca, Formal analysis, Investigation, Visualization, Writing - original draft, Writing - review and editing; Daphne Newell, Alexander Sastokas, Investigation, Visualization, Writing - review and editing; Grant Kinsler, Formal analysis, Visualization, Writing - review and editing

### Author ORCIDs

Kara Schmidlin ⬡ http://orcid.org/0000-0001-6892-0928

Grant Kinsler  https://orcid.org/0000-0001-8308-4665
Kerry Geiler-Samerotte  https://orcid.org/0000-0003-4666-2192

Reviewer #1 (Public review): https://doi.org/10.7554/eLife.94144.3.sa1
Author response https://doi.org/10.7554/eLife.94144.3.sa2

## Additional files

### Supplementary files

• Supplementary file 1. The sequenced mutations present in each of the adaptive lineages on which we performed whole genome sequencing.

• Supplementary file 2. Estimated percentage of diploid cells in each evolution condition at the time point we sampled, determined using nuclear staining and flow cytometry as described in the Methods. *For the High Radicicol +Low Fluconazole condition, 95 instead of 96 samples were measured due to undetectable growth in one well.

• MDAR checklist

### Data availability

All code and data pertaining to this manuscript can be found on Open Science Framework here: https://osf.io/pxyv9/. Whole genomesequences were deposited in GenBank under SRA reference PRJNA1023288.

The following datasets were generated:

| Author(s) | Year | Dataset title | Dataset URL | Database and Identifier |
| --- | --- | --- | --- | --- |
| Schmidlin A, Newell S, Kinsler G | 2024 | Examining fitness tradeoffs across hundreds of drug resistant yeast strains to enumerate different mechanisms of fluconazole resistance | https://osf.io/pxyv9/ | Open Science Framework, pxyv9 |
| Schmidlin A, Newell S, Kinsler G | 2024 | Examining fitness tradeoffs across hundreds of drug resistant yeast strains to enumerate different mechanisms of fluconazole resistance | https://www.ncbi.nlm.nih.gov/bioproject/PRJNA1023288/ | NCBI BioProject, PRJNA1023288 |

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
