## [Editor Report · eLife assessment]

This study provides **valuable** new insights into the trade-offs associated with the evolution of drug resistance in the yeast *S. cerevisiae*, based on a solid approach to evolving and phenotyping hundreds of independent strains. The authors identify distinct phenotypic clusters, defined by their growth across defined conditions, which suggest that tradeoffs are diverse but at the same time could be limited to a few classes according to the underlying resistance mechanisms. The methodologies used align with the current state-of-the-art, and the data and analysis are **solid** as they broadly support the claims, with only a few minor weaknesses remaining after revision. This work will interest molecular biologists working on the evolution of new phenotypes and microbiologists studying multi-drug therapy.

---

## [Referee Report · Reviewer #1 (Public review)]

Summary:

In their manuscript, Schmidlin, Apodaca et al try to answer fundamental questions about the evolution of new phenotypes and the trade-offs associated with this process. As a model, they use yeast resistance to two drugs, fluconazole and radicicol. They use barcoded libraries of isogenic yeasts to evolve thousands of strains in 12 different environments. They then measure the fitness of evolved strains in all environments and use these measurements to enumerate patterns in fitness trade-offs. They identify only six major clusters corresponding to different trade-off profiles, suggesting the vast genotypic landscape of evolved mutants translates to a highly constrained phenotypic space. They sequence over a hundred evolved strains and find that mutations in the same gene can result in different phenotypic profiles.

Overall, the authors deploy innovative methods to scale up experimental evolution experiments, and in many aspects of their approach tried to minimize experimental variation.

Weaknesses:

(1) The main objective of the authors is to characterize the extent of phenotypic diversity in terms of resistance trade-offs between fluconazole and radicicol. To minimize noise in the measurement of relative fitness, the authors only included strains with at least 500 barcode counts across all time points in all 12 experimental conditions, resulting in a set of 774 lineages passing this threshold. As the authors remark, this will bias their datasets for lineages with high fitness in all 12 environments, as all these strains must be fit enough to maintain a high abundance. One of the main observations of the authors is phenotypic space is constrained to a few clusters of roughly similar relative fitness patterns, giving hope that such clusters could be enumerated and considered to design antimicrobial treatment strategies. However, by excluding all lineages that fit in only one or a few environments, they conceal much of the diversity that might exist in terms of trade-offs and set up an inclusion threshold that might present only a small fraction of phenotypic space with characteristics consistent with generalist resistance mechanisms or broadly increased fitness. The general conclusions of the authors regarding the evolution of trade-offs might thus be more focused on multi-drug resistant phenotypes.

(2) Most large-scale pooled competition assays using barcodes are usually stopped after ~25 to avoid noise due to the emergence of secondary mutations. The authors measure fitness across ~40 generations, which is almost the same number of generations as in the evolution experiment. This raises the possibility of secondary mutations biasing abundance values, which would not have been detected by the whole genome sequencing as it was performed before the competition assay. Previous studies approximated the fraction of lineages that could be overtaken by secondary mutations (Venkataram and Dunn et al 2016). In their calculations, Venkataram and Dunn et al defined adaptive mutations in their data as having a selection coefficient of 5% and highly adaptive mutations at around 10%. From this and an estimation of the mutation rate, they estimate that the fraction of lineages overtaken by adaptive mutations is negligible (10^4) after 32 generations. However, the effects on fitness observed by the authors here tend to be much stronger than 5-10%, with relative fitness advantages above 1 and often reaching 2. This could result in a much higher chance of lineages being overtaken at 40 generations.

(3) The approach used by the authors to identify and visualize clusters of phenotypes among lineages does not seem to consider the uncertainty in the measurement of their relative fitness. As can be seen from Figure S4, the inter-replicate difference in measured fitness can often be quite large. From these graphs, it is also possible to see that some of the fitness measurements do not correlate linearly (ex.: Med Flu, Hi Rad Low Flu), meaning that taking the average of both replicates might not be the best approach. Because the clustering approach used does not seem to take this variability into account, it becomes difficult to evaluate the strength of the clustering, especially because the UMAP projection does not include any representation of uncertainty around the position of lineages.

(4) The authors make the decision to use UMAP and a Gaussian mixed model as well as validation data to identify unique clusters, which is one of their main objectives. The choice of 7 clusters as the cutoff for the multiple Gaussian model is not well explained. Based on Figure S6A, BIC starts leveling off at 6 clusters, not 7, and going to 8 clusters would provide the same reduction as going from 6 to 7. This choice also appears arbitrary in Figure S6B, where BIC levels off at 9 clusters when only highly abundant lineages are considered. All of the data presented in the validations is presented to fit within the 6 clusters structure but does not include evidence against alternative scenarios for additional relevant clusters as might be suggested by Figure S6.

(5) Large-scale barcode sequencing assays can often be noisy and are generally validated using growth curves or competition assays. Reconstructing some of the specific mutants they identified to validate their phenotypes would also have been a good addition. If the phenotypic clusters identified cannot be reproduced outside of the sequencing assay, then their relevance are they as a model for multi-drug resistance scenarios might be reduced.

---

## [Author Response]

The following is the authors’ response to the current reviews.

(1) Though we cannot survey all mutants, our observation that 774 genetically diverse adaptive mutants converge at the level of phenotype is important. It adds to growing evidence (see PMID33263280, PMID37437111, PMID22282810, PMID25806684) that the genetic basis of adaptation is not as diverse as the phenotypic basis. This convergence could make evolution more predictable.

(2) Previous fitness competitions using this specific barcode system have been run for greater than 25 generations (PMID33263280, PMID27594428, PMID37861305, PMID27594428). We measure fitness per cycle, rather than per generation, so our fitness advantages are comparable to those in the aforementioned studies, including Venkataram and Dunn et al. (PMID27594428).

(3) Our results remain the same upon removing the ~150 lineages with the noisiest fitness inferences, including those the reviewer mentions (see Figure S7).

(4) We agree that there are likely more than the 6 clusters that we validated with follow-up studies (see Discussion). The important point is that we see a great deal of convergence in the behavior of diverse adaptive mutants.

(5) The growth curves requested by the reviewer were included in our original manuscript; several more were added in the revision (see Figures 5D, 5E, 7D, S11B, S11C).

The following is the authors’ response to the original reviews.

**Public Reviews.**

**Reviewer #1 (Public Review):**
Summary:In their manuscript, Schmidlin, Apodaca, et al try to answer fundamental questions about the evolution of new phenotypes and the trade-offs associated with this process. As a model, they use yeast resistance to two drugs, fluconazole and radicicol. They use barcoded libraries of isogenic yeasts to evolve thousands of strains in 12 different environments. They then measure the fitness of evolved strains in all environments and use these measurements to examine patterns in fitness trade-offs. They identify only six major clusters corresponding to different trade-off profiles, suggesting the vast genotypic landscape of evolved mutants translates to a highly constrained phenotypic space. They sequence over a hundred evolved strains and find that mutations in the same gene can result in different phenotypic profiles.Overall, the authors deploy innovative methods to scale up experimental evolution experiments, and in many aspects of their approach tried to minimize experimental variation.

We thank the reviewer for this positive assessment of our work. We are happy that the reviewer noted what we feel is a unique strength of our approach: we scaled up experimental evolution by using DNA barcodes and by exploring 12 related selection pressures. Despite this scaling up, we still see phenotypic convergence among the 744 adaptive mutants we study.

Weaknesses:

(1) One of the objectives of the authors is to characterize the extent of phenotypic diversity in terms of resistance trade-offs between fluconazole and radicicol. To minimize noise in the measurement of relative fitness, the authors only included strains with at least 500 barcode counts across all time points in all 12 experimental conditions, resulting in a set of 774 lineages passing this threshold. This corresponds to a very small fraction of the starting set of ~21 000 lineages that were combined after experimental evolution for fitness measurements.

This is a misunderstanding that we clarified in this revision. Our starting set did not include 21,000 adaptive lineages. The total number of unique adaptive lineages in this starting set is much lower than 21,000 for two reasons.

First, ~21,000 represents the number of single colonies we isolated in total from our evolution experiments. Many of these isolates possess the same barcode, meaning they are duplicates. Second, and perhaps more importantly, most evolved lineages do not acquire adaptive mutations, meaning that many of the 21,000 isolates are genetically identical to their ancestor. In our revised manuscript, we explicitly stated that these 21,000 isolated lineages do not all represent unique, adaptive lineages. We changed the word “lineages” to “isolates” where relevant in Figure 2 and the accompanying legend. And we have added the following sentence to the figure 2 legend (line 212), “These ~21,000 isolates do not represent as many unique, adaptive lineages because many either have the same barcode or do not possess adaptive mutations.”

More broadly speaking, several previous studies have demonstrated that diverse genetic mutations converge at the level of phenotype and have suggested that this convergence makes adaptation more predictable (PMID33263280, PMID37437111, PMID22282810, PMID25806684). Most of these studies survey fewer than 774 mutants. Further, our study captures mutants that are overlooked in previous studies, such as those that emerge across subtly different selection pressures (e.g., 4 𝜇g/ml vs. 8 𝜇g/ml flu) and those that are undetectable in evolutions lacking DNA barcodes. Thus, while our experimental design misses some mutants (see next comment), it captures many others. Thus, we feel that “our work – showing that 774 mutants fall into a much smaller number of groups” is important because it “contributes to growing literature suggesting that the phenotypic basis of adaptation is not as diverse as the genetic basis (lines 176 - 178).”

As the authors briefly remark, this will bias their datasets for lineages with high fitness in all 12 environments, as all these strains must be fit enough to maintain a high abundance.

We now devote 19 lines of text to discussing this bias (on lines 160 - 162, 278-284, and in more detail on 758 - 767).

We walk through an example of a class of mutants that our study misses. One lines 759 - 763, we say, “our study is underpowered to detect adaptive lineages that have low fitness in any of the 12 environments. This is bound to exclude large numbers of adaptive mutants. For example, previous work has shown some FLU resistant mutants have strong tradeoffs in RAD (Cowen and Lindquist 2005). Perhaps we are unable to detect these mutants because their barcodes are at too low a frequency in RAD environments, thus they are excluded from our collection of 774.”

In our revised version, we added more text earlier in the manuscript that explicitly discusses this bias. Lines 278 – 283 now read, “The 774 lineages we focus on are biased towards those that are reproducibly adaptive in multiple environments we study. This is because lineages that have low fitness in a particular environment are rarely observed >500 times in that environment (Figure S4). By requiring lineages to have high-coverage fitness measurements in all 12 conditions, we may be excluding adaptive mutants that have severe tradeoffs in one or more environments, consequently blinding ourselves to mutants that act via unique underlying mechanisms.”

Note that while we “miss” some classes of mutants, we “catch” other classes that may have been missed in previous studies of convergence. For example, we observe a unique class of FLU-resistant mutants that primarily emerged in evolution experiments that lack FLU (Figure 3). Thus, we think that the unique design of our study, surveying 12 environments, allows us to make a novel contribution to the study of phenotypic convergence.

One of the main observations of the authors is phenotypic space is constrained to a few clusters of roughly similar relative fitness patterns, giving hope that such clusters could be enumerated and considered to design antimicrobial treatment strategies. However, by excluding all lineages that fit in only one or a few environments, they conceal much of the diversity that might exist in terms of trade-offs and set up an inclusion threshold that might present only a small fraction of phenotypic space with characteristics consistent with generalist resistance mechanisms or broadly increased fitness. This has important implications regarding the general conclusions of the authors regarding the evolution of trade-offs.

We agree and discussed exactly the reviewer’s point about our inclusion threshold in the 19 lines of text mentioned previously (lines 160 - 162, 278-284, and 758 - 767). To add to this discussion, and avoid the misunderstanding the reviewer mentions, we added the following strongly-worded sentence to the end of the paragraph on lines 749 – 767 in our revised manuscript: “This could complicate (or even make impossible) endeavors to design antimicrobial treatment strategies that thwart resistance”.

More generally speaking, we set up our study around Figure 1, which depicts a treatment strategy that works best if there exists but a single type of adaptive mutant. Despite our inclusion threshold, we find there are at least 6 types of mutants. This diminishes hopes of designing simple multidrug strategies like Figure 1. Our goal is to present a tempered and nuanced discussion of whether and how to move forward with designing multidrug strategies, given our observations. On one hand, we point out how the phenotypic convergence we observe is promising. But on the other hand, we also point out how there may be less convergence than meets the eye for various reasons including the inclusion threshold the reviewer mentions (lines 749 - 767).

We have made several minor edits to the text with the goal of providing a more balanced discussion of both sides. For example, we added the words, “may yet” to the following sentences on lines 32 – 36 of the abstract: “These findings, on one hand, demonstrate the difficulty in relying on consistent or intuitive tradeoffs when designing multidrug treatments. On the other hand, by demonstrating that hundreds of adaptive mutations can be reduced to a few groups with characteristic tradeoffs, our findings may yet empower multidrug strategies that leverage tradeoffs to combat resistance.”

(2) Most large-scale pooled competition assays using barcodes are usually stopped after ~25 to avoid noise due to the emergence of secondary mutations.

The rate at which new mutations enter a population is driven by various factors such as the mutation rate and population size, so choosing an arbitrary threshold like 25 generations is difficult.

We conducted our fitness competition following previous work using the Levy/Blundell yeast barcode system, in which the number of generations reported varies from 32 to 40 (PMID33263280, PMID27594428, PMID37861305, see PMID27594428 for detailed calculation of the fraction of lineages biased by secondary mutations in this system).

The authors measure fitness across ~40 generations, which is almost the same number of generations as in the evolution experiment. This raises the possibility of secondary mutations biasing abundance values, which would not have been detected by the whole genome sequencing as it was performed before the competition assay.

Previous work has demonstrated that in this evolution platform, most mutations occur during the transformation that introduces the DNA barcodes (Levy et al. 2015). In other words, these mutations are already present and do not accumulate during the 40 generations of evolution. Therefore, the observation that we collect a genetically diverse pool of adaptive mutants after 40 generations of evolution is not evidence that 40 generations is enough time for secondary mutations to bias abundance values.

We have added the following sentence to the main text to highlight this issue (lines 247 - 249): “This happens because the barcoding process is slightly mutagenic, thus there is less need to wait for DNA replication errors to introduce mutations (Levy et al. 2015; Venkataram et al. 2016).”

We also elaborate on this in the method section entitled, “Performing barcoded fitness competition experiments,” where we added a full paragraph to clarify this issue (lines 972 - 980).

(3) The approach used by the authors to identify and visualize clusters of phenotypes among lineages does not seem to consider the uncertainty in the measurement of their relative fitness. As can be seen from Figure S4, the inter-replicate difference in measured fitness can often be quite large. From these graphs, it is also possible to see that some of the fitness measurements do not correlate linearly (ex.: Med Flu, Hi Rad Low Flu), meaning that taking the average of both replicates might not be the best approach. Because the clustering approach used does not seem to take this variability into account, it becomes difficult to evaluate the strength of the clustering, especially because the UMAP projection does not include any representation of uncertainty around the position of lineages. This might paint a misleading picture where clusters appear well separate and well defined but are in fact much fuzzier, which would impact the conclusion that the phenotypic space is constricted.

Our noisiest fitness measurements correspond to barcodes that are the least abundant and thus suffer the most from stochastic sampling noise. These are also the barcodes that introduce the nonlinearity the reviewer mentions. We removed these from our dataset by increasing our coverage threshold from 500 reads to 5,000 reads. The clusters did not collapse, which suggests that they were not capturing this noise (Figure S7B).

More importantly, we devoted 4 figures and 200 lines of text to demonstrating that the clusters we identified capture biologically meaningful differences between mutants (and not noise). We have modified the main text to point readers to figures 5 through 8 earlier, such that it is more apparent that the clustering analysis is just the first piece of our data demonstrating convergence at the level of phenotype.

(4) The authors make the decision to use UMAP and a gaussian mixed model to cluster and represent the different fitness landscapes of their lineages of interest. Their approach has many caveats. First, compared to PCA, the axis does not provide any information about the actual dissimilarities between clusters. Using PCA would have allowed a better understanding of the amount of variance explained by components that separate clusters, as well as more interpretable components.

The components derived from PCA are often not interpretable. It’s not obvious that each one, or even the first one, will represent an intuitive phenotype, like resistance to fluconazole. Moreover, we see many non-linearities in our data. For example, fitness in a double drug environment is not predicted by adding up fitness in the relevant single drug environments. Also, there are mutants that have high fitness when fluconazole is absent or abundant, but low fitness when mild concentrations are present. These types of nonlinearities can make the axes in PCA very difficult to interpret, plus these nonlinearities can be missed by PCA, thus we prefer other clustering methods.

Still, we agree that confirming our clusters are robust to different clustering methods is helpful. We have included PCA in the revised manuscript, plotting PC1 vs PC2 as Figure S9 with points colored according to the cluster assignment in figure 4 (i.e. using a gaussian mixture model). It appears the clusters are largely preserved.

Second, the advantages of dimensional reduction are not clear. In the competition experiment, 11/12 conditions (all but the no drug, no DMSO conditions) can be mapped to only three dimensions: concentration of fluconazole, concentration of radicicol, and relative fitness. Each lineage would have its own fitness landscape as defined by the plane formed by relative fitness values in this space, which can then be examined and compared between lineages.

We worry that the idea stems from apriori notions of what the important dimensions should be. The biology of our system is unfortunately not intuitive. For example, it seems like this idea would miss important nonlinearities such as our observation that low fluconazole behaves more like a novel selection pressure than a dialed down version of high fluconazole.

Third, the choice of 7 clusters as the cutoff for the multiple Gaussian model is not well explained. Based on Figure S6A, BIC starts leveling off at 6 clusters, not 7, and going to 8 clusters would provide the same reduction as going from 6 to 7. This choice also appears arbitrary in Figure S6B, where BIC levels off at 9 clusters when only highly abundant lineages are considered.

We agree. We did not rely on the results of BIC alone to make final decisions about how many clusters to include. Another factor we considered were follow-up genotyping and phenotyping studies that confirm biologically meaningful differences between the mutants in each cluster (Figures 5 – 8). We now state this explicitly. Here is the modified paragraph where we describe how we chose a model with 7 clusters, from lines 436 – 446 of the revised manuscript:

“Beyond the obvious divide between the top and bottom clusters of mutants on the UMAP, we used a gaussian mixture model (GMM) (Fraley and Raftery, 2003) to identify clusters. A common problem in this type of analysis is the risk of dividing the data into clusters based on variation that represents measurement noise rather than reproducible differences between mutants (Mirkin, 2011; Zhao et al., 2008). One way we avoided this was by using a GMM quality control metric (BIC score) to establish how splitting out additional clusters affected model performance (Figure S6). Another factor we considered were follow-up genotyping and phenotyping studies that demonstrate biologically meaningful differences between mutants in different clusters (Figures 5 – 8). Using this information, we identified seven clusters of distinct mutants, including one pertaining to the control strains, and six others pertaining to presumed different classes of adaptive mutant (Figure 4D). It is possible that there exist additional clusters, beyond those we are able to tease apart in this study.”

This directly contradicts the statement in the main text that clusters are robust to noise, as more a stringent inclusion threshold appears to increase and not decrease the optimal number of clusters. Additional criteria to BIC could have been used to help choose the optimal number of clusters or even if mixed Gaussian modeling is appropriate for this dataset.

We are under the following impression: If our clustering method was overfitting, i.e. capturing noise, the optimal number of clusters should decrease when we eliminate noise. It increased. In other words, the observation that our clusters did not collapse (i.e. merge) when we removed noise suggests these clusters were not capturing noise.

Most importantly, our validation experiments, described below, provide additional evidence that our clusters capture meaningful differences between mutants (and not noise).

(5) Large-scale barcode sequencing assays can often be noisy and are generally validated using growth curves or competition assays.

Some types of bar-seq methods, in particular those that look at fold change across two time points, are noisier than others that look at how frequency changes across multiple timepoints (PMID30391162). Here, we use the less noisy method. We also reduce noise by using a stricter coverage threshold than previous work (e.g., PMID33263280), and by excluding batch effects by performing all experiments simultaneously, since we found this to be effective in our previous work (PMID37237236).

Perhaps also relevant is that the main assay we use to measure fitness has been previously validated (PMID27594428) and no subsequent study using this assay validates using the methods suggested above (see PMID37861305, PMID33263280, PMID31611676, PMID29429618, PMID37192196, PMID34465770, PMID33493203). Similarly, bar-seq has been used, without the suggested validation, to demonstrate that the way some mutant’s fitness changes across environments is different from other mutants (PMID33263280, PMID37861305, PMID31611676, PMID33493203, PMID34596043). This is the same thing that we use bar-seq to demonstrate.

For all of these reasons above, we are hesitant to confirm bar-seq itself as a valid way to infer fitness. It seems this is already accepted as a standard in our field. However, please see below.

Having these types of results would help support the accuracy of the main assay in the manuscript and thus better support the claims of the authors.

While we don’t agree that fitness measurements obtained from this bar-seq assay generally require validation, we do agree that it is important to validate whether the mutants in each of our 6 clusters indeed are different from one another in meaningful ways.

Our manuscript has 4 figures (5 - 8) and over 200 lines of text dedicated to validating whether our clusters capture reproducible and biologically meaningful differences between mutants. In the revised manuscript, we added additional validation experiments, such that three figures (Figures 5, 7 and S11) now involve growth curves, as the reviewer requested.

Below, we walk through the different types of validation experiments that are present in our manuscript, including those that were added in this revision.

(1) Mutants from different clusters have different growth curves: In our original manuscript, we measured growth curves corresponding to a fitness tradeoff that we thought was surprising. Mutants in clusters 4 and 5 both have fitness advantages in single drug conditions. While mutants from cluster 4 also are advantageous in the relevant double drug conditions, mutants from cluster 5 are not! We validated these different behaviors by studying growth curves for a mutant from each cluster (Figures 7 and S11), finding that mutants from different clusters have different growth curves. In the revised manuscript, we added growth curves for 6 additional mutants (3 from cluster 1 and 3 from cluster 3), demonstrating that only the cluster 1 mutants have a tradeoff in high concentrations of fluconazole (see Figure 5D & 5E). In sum, this work demonstrates that mutants from different clusters have predictable differences in their growth phenotypes.

(2) Mutants from different clusters have different evolutionary origins: In our original manuscript, we came up with a novel way to ask whether the clusters capture different types of adaptive mutants. We asked whether the mutants in each cluster originate from different evolution experiments. They often do (see pie charts in Figures 5, 6, 7, 8). In the revised manuscript, we extended this analysis to include mutants from cluster 1. Cluster 1 is defined by high fitness in low fluconazole that declines with increasing fluconazole. In our revised manuscript, we show that cluster 1 lineages were overwhelmingly sampled from evolutions conducted in our lowest concentration of fluconazole (see pie chart in new Figure 5A). No other cluster’s evolutionary history shows this pattern (compare to pie charts in figures 6, 7, and 8).

**These pie charts also provide independent confirmation supporting the fitness tradeoffs observed for each cluster in figure 4E. For example, mutants in cluster 5 appear to have a tradeoff in a particular double drug condition (HRLF), and the pie charts confirm that they rarely originate from that evolution condition. This differs from cluster 4 mutants, which do not have a fitness tradeoff in HRLF, and are more likely to originate from that environment (see purple pie slice in figure 7). Additional cases where results of evolution experiments (pie charts) confirm observed fitness tradeoffs are discussed in the manuscript on lines 320 – 326, 594 – 598, 681 – 685.

(3) Mutants from each cluster often fall into different genes: We sequenced many of these mutants and show that mutants in the same gene are often found in the same cluster. For example, all 3 IRA1 mutants are in cluster 6 (Fig 8), both GPB2 mutants are in cluster 4 (Figs 7 & 8), and 35/36 PDR mutants are in either cluster 2 or 3 (Figs 5 & 6).

(4) Mutants from each cluster have behaviors previously observed in the literature: We compared our sequencing results to the literature and found congruence. For example, PDR mutants are known to provide a fitness benefit in fluconazole and are found in clusters that have high fitness in fluconazole (lines 485 - 491). Previous work suggests that some mutations to PDR have different tradeoffs than others, which corresponds to our finding that PDR mutants fall into two separate clusters (lines 610 - 612). IRA1 mutants were previously observed to have high fitness in our “no drug” condition and are found in the cluster that has the highest fitness in the “no drug” condition (lines 691 - 696). Previous work even confirms the unusual fitness tradeoff we observe where IRA1 and other cluster 6 mutants have low fitness only in low concentrations of fluconazole (lines 702 - 704).

(5) Mutants largely remain in their clusters when we use alternate clustering methods: In our original manuscript, we performed various different re-clustering and/or normalization approaches on our data (Fig 6, S5, S7, S8, S10). The clusters of mutants that we observe in figure 4 do not change substantially when we re-cluster the data. In our revised manuscript, we added another clustering method: principal component analysis (PCA) (Fig S9). Again, we found that our clusters are largely preserved.

While these experiments demonstrate meaningful differences between the mutants in each cluster, important questions remain. For example, a long-standing question in biology centers on the extent to which every mutation has unique phenotypic effects versus the extent to which scientists can predict the effects of some mutations from other similar mutations. Additional studies on the clusters of mutants discovered here will be useful in deepening our understanding of this topic and more generally of the degree of pleiotropy in the genotype-phenotype map.

**Reviewer #2 (Public Review):**
Summary:Schmidlin & Apodaca et al. aim to distinguish mutants that resist drugs via different mechanisms by examining fitness tradeoffs across hundreds of fluconazole-resistant yeast strains. They barcoded a collection of fluconazole-resistant isolates and evolved them in different environments with a view to having relevance for evolutionary theory, medicine, and genotypephenotype mapping.Strengths:There are multiple strengths to this paper, the first of which is pointing out how much work has gone into it; the quality of the experiments (the thought process, the data, the figures) is excellent. Here, the authors seek to induce mutations in multiple environments, which is a really large-scale task. I particularly like the attention paid to isolates with are resistant to low concentrations of FLU. So often these are overlooked in favour of those conferring MIC values >64/128 etc. What was seen is different genotype and fitness profiles. I think there's a wealth of information here that will actually be of interest to more than just the fields mentioned (evolutionary medicine/theory).

We are grateful for this positive review. This was indeed a lot of work! We are happy that the reviewer noted what we feel is a unique strength of our manuscript: that we survey adaptive isolates across multiple environments, including low drug concentrations.

Weaknesses:Not picking up low fitness lineages - which the authors discuss and provide a rationale as to why. I can completely see how this has occurred during this research, and whilst it is a shame I do not think this takes away from the findings of this paper. Maybe in the next one!

We thank the reviewer for these words of encouragement and will work towards catching more low fitness lineages in our next project.

In the abstract the authors focus on 'tradeoffs' yet in the discussion they say the purpose of the study is to see how many different mechanisms of FLU resistance may exist (lines 679-680), followed up by "We distinguish mutants that likely act via different mechanisms by identifying those with different fitness tradeoffs across 12 environments". Whilst I do see their point, and this is entirely feasible, I would like a bit more explanation around this (perhaps in the intro) to help lay-readers make this jump. The remainder of my comments on 'weaknesses' are relatively fixable, I think:

We have expanded the introduction, in particular lines 129 – 157 of the revised manuscript, to walk readers through the connection between fitness tradeoffs and molecular mechanisms. For example, here is one relevant section of new text from lines 131 - 136: “The intuition here is as follows. If two groups of drug resistant mutants have different fitness tradeoffs, it could mean that they provide resistance through different underlying mechanisms. Alternatively, both could provide drug resistance via the same mechanism, but some mutations might also affect fitness via additional mechanisms (i.e. they might have unique “side-effects” at the molecular level) resulting in unique fitness tradeoffs in some environments.”

In the introduction I struggle to see how this body of research fits in with the current literature, as the literature cited is a hodge-podge of bacterial and fungal evolution studies, which are very different! So example, the authors state "previous work suggests that mutants with different fitness tradeoffs may affect fitness through different molecular mechanisms" (lines 129-131) and then cite three papers, only one of which is a fungal research output. However, the next sentence focuses solely on literature from fungal research. Citing bacterial work as a foundation is fine, but as you're using yeast for this I think tailoring the introduction more to what is and isn't known in fungi would be more appropriate. It would also be great to then circle back around and mention monotherapy vs combination drug therapy for fungal infections as a rationale for this study. The study seems to be focused on FLU-resistant mutants, which is the first-line drug of choice, but many (yeast) infections have acquired resistance to this and combination therapy is the norm.

We ourselves are broadly interested in the structure of the genotype-phenotype-fitness map (PMID33263280, PMID32804946). For example, we are interested in whether diverse mutations converge at the level of phenotype and fitness. Figure 1A depicts a scenario with a lot of convergence in that all adaptive mutations have the same fitness tradeoffs.

The reason we cite papers from yeast, as well as bacteria and cancer, is that we believe general conclusions about the structure of the genotype-phenotype-fitness map apply broadly. For example, the sentence the reviewer highlights, “previous work suggests that mutants with different fitness tradeoffs may affect fitness through different molecular mechanisms” is a general observation about the way genotype maps to fitness. So, we cited papers from across the tree of life to support this sentence. And in the next sentence, where we cite 3 papers focusing solely on fungal research, we cite them because they are studies about the complexity of this map. Their conclusions, in theory, should also apply broadly, beyond yeast.

On the other hand, because we study drug resistant mutations, we hope that our dataset and observations are of use to scientists studying the evolution of resistance. We use our introduction to explain how the structure of the genotype-phenotype-fitness map might influence whether a multidrug strategy is successful (Figure 1).

We are hesitant to rework our introduction to focus more specifically on fungal infections as this is not our primary area of expertise.

Methods: Line 769 - which yeast? I haven't even seen mention of which species is being used in this study; different yeast employ different mechanisms of adaptation for resistance, so could greatly impact the results seen. This could help with some background context if the species is mentioned (although I assume *S. cerevisiae*).

In the revised manuscript, we have edited several lines (line 95, 186, 822) to state the organism this work was done with is *Saccharomyces cerevisiae*.

In which case, should aneuploidy be considered as a mechanism? This is mentioned briefly on line 556, but with all the sequencing data acquired this could be checked quickly?

We like this idea and we are working on it, but it is not straightforward. The reviewer is correct in that we can use the sequencing data that we already have. But calling aneuploidy with certainty is tough because its signal can be masked by noise. In other words, some regions of the genome may be sequenced more than others by chance.

Given this is not straightforward, at least not for us, this analysis will likely have to wait for a subsequent paper.

I think the authors could be bolder and try and link this to other (pathogenic) yeasts. What are the implications of this work on say, Candida infections?

Perhaps because our background lies in general study of the genotype-phenotype map, we are hesitant about making bold assertions about how our work might apply to pathogenic yeasts. We are hopeful that our work will serve as a stepping-stone such that scientists from that community can perhaps make (and test) such statements.

**Recommendations for the authors:**

**Reviewer #1 (Recommendations For The Authors):**
I found the ideas and the questions asked in this manuscript to be interesting and ambitious. The setup of the evolution and fitness competition experiments was well poised to answer them, but the analysis of the data is not currently enough to properly support the claims made. I would suggest revising the analysis to address the weaknesses raised in the public review and if possible, adding some more experimental validations. As you already have genome sequencing data showing the causal mutation for many mutants across the different clusters, it should be possible for you to reconstruct some of the strains and test validate their phenotypes and cluster identity.

Yes, this is possible. We added more validation experiments (see figure 5). We already had quite a few validation experiments (figures 5 - 8 and lines 479 - 718), but we did not clearly highlight the significance of these analyses in our original manuscript. Therefore, we modified the text in our revised manuscript in various places to do so. For example, we now make clearer that we jointly use BIC scores as well as validation experiments to decide how many clusters to describe (lines 436 - 446). We also make clearer that our clustering analysis is only the first step towards identifying groups of mutants with similar tradeoffs by using words and phrases like, “we start by” (line 411) and “preliminarily” (line 448) when discussing the clustering analysis. We also point readers to all the figures describing our validation experiments earlier (line 443), and list these experiments out in the discussion (lines 738 - 741).

Also, please deposit your genome sequencing data in a public database (I am not sure I saw it mentioned anywhere).

We have updated line 1088 of the methods section to include this sentence: “Whole genome sequences were deposited in GenBank under SRA reference PRJNA1023288.”

**Reviewer #2 (Recommendations For The Authors):**
I don't think the figures or experiments can be improved upon, they are excellent. There are a few times I feel things are written in a rather confusing way and could be explained better, but also I feel there are places the authors jump from one thing to another really quickly and the reader (who might not be an expert in this area) will struggle to keep up. For example:Explaining what RAD is - it is introduced in the methods, but what it is, is not really explained.

Since the introduction is already very long, we chose not to explain radicicol’s mechanism of action here. Instead, we bring this up later on lines 614 – 621 when it becomes relevant.

More generally, in response to this advice and that from reviewer 1, we also added text to various places in the manuscript to help explain our work more clearly. In particular, we clarified the significance of our validation experiments and various important methodological details (see above). We also better explained the connection between fitness tradeoffs and mechanisms (see above) and added more details about the potential use cases of our approach (lines 142 – 150).

The abstract states "some of the groupings we find are surprising. For example, we find some mutants that resist single drugs do not resist their combination, and some mutants to the same gene have different tradeoffs than others". Firstly, this sentence is a bit confusing to read but if I've read it as intended, then is it really surprising? It's difficult for organisms (bacteria and fungi) to develop multiple beneficial mutations conferring drug resistance on the same background, hence why combination antifungal drug therapy is often used to treat infections.

This is a place where brevity got in the way of clarity. We added a bit of text to make clear why we were surprised. Specifically, we were surprised because not all mutants behave the same. Some resist single drugs AND their combination. Some resist single drugs but not their combination. The sentence in the abstract now reads, “For example, we find some mutants that resist single drugs do not resist their combination, while others do. And some mutants to the same gene have different tradeoffs than others.”